# AMPLITUDE-BASED INPUT ATTRIBUTION IN QUANTUM LEARNING VIA INTEGRATED GRADIENTS

## ABSTRACT

Quantum machine learning (QML) algorithms have demonstrated early promise across hardware platforms, but remain difficult to interpret due to the inherent opacity of quantum state evolution. Widely used classical interpretability methods, such as integrated gradients and surrogate-based sensitivity analysis, are not directly compatible with quantum circuits due to measurement collapse and the exponential complexity of simulating state evolution. In this work, we introduce HATTRIQ, a general-purpose framework to compute amplitude-based input attribution scores in circuit-based QML models. HATTRIQ supports the widely-used input amplitude embedding feature encoding scheme and uses a Hadamard test–based construction to compute input gradients directly on quantum hardware to generate provably faithful attributions. We validate HATTRIQ on classification tasks across several datasets (Bars and Stripes, MNIST, and FashionMNIST).

## 1 INTRODUCTION

Quantum machine learning (QML) uses quantum computing to enhance data analysis and pattern recognition in AI. By using quantum features like superposition and entanglement, QML algorithms have the potential to offer speedups over classical methods (Biamonte et al., 2017; DeRieux & Saad, 2025; De La Vega et al., 2023). Current research emphasizes hybrid models, where quantum circuits work alongside classical optimizers (Bharti et al., 2022; Arrasmith et al., 2021), with applications in classification, clustering, and generative tasks (Preskill, 2018; DiBrita et al., 2024; Zhang et al., 2023; Han et al., 2025). While limited by today's hardware, QML holds promise for solving complex problems in fields such as healthcare, finance, and scientific computing as quantum systems advance (Nicoli et al., 2023; Hothem et al., 2024; Preskill, 2023; Cerezo et al., 2022). Despite growing interest and experimental progress, QML models remain difficult to interpret due to the inherent opacity of quantum state evolution and the absence of intermediate observability mid computation (Herbst et al., 2025; Pira & Ferrie, 2024; Heese et al., 2025).

In classical machine learning, interpretability methods such as feature attribution play a critical role in understanding model predictions, particularly in sensitive and mission-critical domains like healthcare and autonomous systems (Radenovic et al., 2022; Zimmermann et al., 2023; Agarwal et al., 2021; Hooker et al., 2019; Alvarez Melis & Jaakkola, 2018). Attribution methods (Rudin, 2019; Krishna et al., 2022) – such as integrated gradients (Sundararajan et al., 2017) – assign importance scores to input features, revealing which aspects of the input most influence the model's output. These methods enhance transparency, support debugging, and build trust in model behavior. In contrast, existing QML pipelines provide little insight into how input features affect final measurement outcomes, especially when data is encoded and compressed into high-dimensional quantum state amplitudes (Jerbi et al., 2021; Bausch, 2020; Preskill, 2018; 2023).

In this work, we propose HATTRIQ[1], a methodology for computing the input attribution scores for quantum circuits. HATTRIQ adapts integrated gradients (Sundararajan et al., 2017) to the quantum circuit setting, enabling attribution for amplitude embedding. Leveraging integrated gradients for quantum models is challenging, as larger models require working in exponentially large Hilbert spaces and manipulating complex amplitude vectors, making both analysis and simulation resource-intensive (Xiong et al., 2024; Lei et al., 2024). Another challenge is that quantum states are hidden

---

[1]HATTRIQ stands for Hadamard test-based input attribution score scheme for quantum models.

from the user during computation. For large programs, we cannot simply record or log the hidden state after each circuit layer, as any attempt to measure the hidden state collapses the quantum state of the circuit entirely (Gong & Aaronson, 2023; Abbas et al., 2023); traditional (surrogate-based) sensitivity and Sobol/Shapley score methods (Owen, 2014; Cho et al., 2025) cannot preserve unitarity in quantum circuits, making it difficult to understand how different signals are propagated through the computation circuit.

To address this, HATTRIQ implements a Hadamard test–based construction that computes exact gradients directly on quantum hardware, without requiring access to internal quantum states. For fault-tolerant quantum devices, where the impact of hardware noise (Akhalwaya et al., 2024; Wu et al., 2025; Patel et al., 2024) is negligible, we propose a parallelization mechanism to evaluate multiple gradient components concurrently. In doing so, HATTRIQ enables input-level sensitivity analysis for amplitude-encoded data, which classical IG cannot provide efficiently. Our main result yields a hardware-compatible way to access the necessary derivatives for IG calculation, without knowing the internal quantum state of the the computer. Practically, this enables the identification of highly influential features, and to check if these align with semantically meaningful regions in its prediction. It also allows us to comparing encodings with similar accuracy but different attribution patterns, and revealing bias toward background or artifacts.

**Our contributions are as follows.**

- We introduce a formalism for computing integrated gradients in QML models that utilize amplitude embedding for input encoding; the formalism also works with other encodings such as angle embedding.

- We present a quantum-native circuit construction based on the Hadamard test to compute exact feature gradients for amplitude-embedded input attribution.

- We provide a multi-ancilla-based parallelization technique to enable gradient computation concurrently on larger quantum devices with sufficient capacity.

- We evaluate HATTRIQ on multiple classification tasks across Bars and Stripes (Bowles et al., 2024), MNIST (LeCun, 1998), and FashionMNIST (Xiao et al., 2017) datasets, demonstrating high-fidelity attribution.

## 2 RELEVANT CONCEPTS

### 2.1 QUANTUM STATES AND GATES

Quantum computations are performed by quantum circuits, which manipulate qubits with logic gates. The *state* of a qubit is represented as a vector: $|\psi\rangle = \beta_0 |0\rangle + \beta_1 |1\rangle$, where $\beta_i$ is a complex coefficient for basis state $|i\rangle$. The probability of measuring the qubit to be in state $|i\rangle$ is $|\beta_i|^2$, which means we must have $|\beta_0|^2 + |\beta_1|^2 = 1$ (Schuld & Killoran, 2019; Silver et al., 2023). For an $n$ qubit system, the statevector is a complex vector $|\psi\rangle \in \mathbb{C}^{2^n}$ that is normalized $\langle\psi|\psi\rangle = 1$. States are then written in terms of an orthonormal basis set; the conventional choice is referred to as the computational basis set. If we define $b_k$ as the bitstring corresponding to integer $k$, we can define the computational basis as the set $\{|b_k\rangle \; \forall \, k \in \mathbb{Z}, 0 \leq k \leq 2^n - 1\}$. Our state can then be expressed as $|\psi\rangle = \sum_{k=0}^{2^n-1} \beta_k |b_k\rangle$ (Schuld & Killoran, 2019; Silver et al., 2023). Logic gates are represented by unitary matrices ($U$) acting on states: $U |\psi_1\rangle = |\psi_2\rangle$. Circuits are constructed by composing sequences of gates together (White et al., 2001; Srinivasan et al., 2018).

### 2.2 PARAMETERIZED QUANTUM CIRCUITS

We study quantum models that can be represented by unitary circuits $U$ and measured observables $O$. While our technique is broadly applicable to general classes of QML models, for demonstration purposes, we focus on circuits with trainable gate parameters. These parameterized quantum circuits (PQCs) are also referred to as variational quantum circuits and have found extensive applications in quantum machine learning, quantum chemistry, and other areas of quantum optimization (Bharti et al., 2022; Arrasmith et al., 2021). Often, the trainable gates in PQCs are rotation gates, which rotate the quantum state according to some angle parameter. There are many possible ways to arrange

a PQC; the fixed structure of a PQC is referred to as an ansatz, and is analogous to fixing a neural network architecture.

Let $\mathbf{x} \in \mathbb{R}^D$ be a data point, and $V(\mathbf{x}) |0\rangle = |x\rangle \in \mathbb{C}^{2^n}$ be a the quantum state that encodes it, with $V(\mathbf{x})$ being the circuit that performs the encoding. Let $U(\boldsymbol{\theta})$ be a PQC with trainable parameters $\boldsymbol{\theta}$ (Schleich et al., 2024), and $O$ be a Hermitian operator that represents the observable measured for the model output. We consider quantum models which apply some circuit operations to the input state $|x\rangle$ and then compute an expectation value, written as

$$F(\mathbf{x} ; \boldsymbol{\theta}) = \langle x | U^\dagger(\boldsymbol{\theta}) \, O \, U(\boldsymbol{\theta}) \, |x\rangle . \tag{1}$$

In the more general case, we might compose $F(\mathbf{x} ; \boldsymbol{\theta})$ with some other (likely nonlinear) function to add complexity to our model: our discussion generalizes simply by applying the chain rule (Arrasmith et al., 2021) in the gradient computation as introduced next. The same is true for hybrid quantum-classical models, without substantial change to the methodology. In hybrid architectures, a quantum layer with amplitude encoding can be treated as a differentiable block: HATTRIQ supplies its input gradient, which can then be combined with classical IG in preceding layers via the chain rule.

*Remark* 2.1. For our discussion, we do not place any specific requirements on $U$, except that it must be a valid unitary operator. In most applications, however, $U$ will have some fixed structure of gates (ansatz). Some subset of these gates will depend on variational parameters $\boldsymbol{\theta}$, which are then optimized to minimize the loss. Later, we will also require that observable $O$ be unitary as well as Hermitian, as is the case with the standard Pauli operators.

## 2.3 INTEGRATED GRADIENTS

We base our technique on the integrated gradients method proposed in (Sundararajan et al., 2017). This work studies the problem of attributing the prediction of deep learning networks to input features in a sample. Integrate gradients benefit from an axiomatic formulation, with guarantees about their sensitivity and implementation invariance (Sundararajan et al., 2017; Mudrakarta et al., 2018).

In addition to its superior theoretical properties, this method for attribution also only relies on a small number of model evaluations and gradient computations, without the need for additional knowledge of the hidden state (Sundararajan et al., 2017). This is highly desirable for the quantum setting, where measuring and storing the internal state at multiple points during the computation would incur a high overhead.

**Definition 2.2** (Attribution Score). *The integrated gradients attribution of a sample* $\mathbf{x}$ *relative to baseline* $\mathbf{x}'$ *is given as the following integral:*

$$IG_i(x) = (x_i - x_i') \int_0^1 \frac{\partial F(x' + \alpha \cdot (x - x'))}{\partial x_i} d\alpha. \tag{2}$$

*The calculated value* $IG_i$ *is the integrated gradients attribution for the* $i^{th}$ *feature, and it represents the contribution that it makes to the final model prediction.*

## 3 FEATURE GRADIENTS

In this section, we introduce the most popular schemes for encoding data features into a quantum circuit calculation: (1) angle embedding and (2) amplitude embedding (Havlíček et al., 2019; Schuld & Petruccione, 2018; Lloyd et al., 2020; Iten et al., 2016; Schuld & Killoran, 2019). For each of these encoding methods, we introduce our methodology for computing the gradients with respect to those encoded features, attributing the circuit output to features.

## 3.1 ANGLE EMBEDDING (OR ENCODING)

For angle-embedded data, the preparation circuit $V(\mathbf{x})$ consists of rotation gates, $\{R(x_i)\}$ each of which depends on an angle parameter. The angle parameters used are the features $x_i$. In such cases, the gradient with respect to the features can be natively calculated using the well-known parameter shift rule (Mitarai et al., 2018; Schuld et al., 2019), which allows for computing the gradient of quantum circuits by re-executing those circuits with shifted parameter values. For a quantum gate parametrized by $\theta_i$ and with only two distinct eigenvalues $\pm r$, it has been shown (Schuld et al., 2019):

$$\frac{\partial F}{\partial \theta_i} = r \left[ F(\theta_i + s) - F(\theta_i - s) \right] \tag{3}$$

where $s = \frac{\pi}{4r}$ is the required shift. While at first glance this formula is reminiscent of a standard finite difference, it differs in that the shift $s$ is not taken to be infinitesimal, and the result of this calculation is exact. This requires two additional circuit evaluations per parameter, making the gradient calculation linear with respect to the number of parameters. While Eq. 3 is not generally applicable to all gates, many parameterized gates, like single qubit rotations, do satisfy the eigenvalue requirements, and parameter shift has been utilized in a variety of quantum optimization settings (Schuld et al., 2020; Arrasmith et al., 2021). Additional rules have been formulated that generalize this result to additional kinds of parameterized gates (Wierichs et al., 2022).

While its simplicity makes angle embedding an attractive choice for near-term applications, the number of encoded features typically grows only linearly with the number of qubits (Schuld et al., 2020), meaning the angle encoding does not make full use of the exponentially large Hilbert space, and does not reach the information upper bound on a sphere (Luo et al., 2024).

### 3.2 AMPLITUDE EMBEDDING (OR ENCODING)

In the amplitude embedding case (Khan et al., 2024), data features are encoded as amplitudes of the input state: $|x\rangle = \sum_i x_i |b_i\rangle$. Unlike this angle embedding case, this allows for encoding exponentially many input features relative to the number of qubits, expanding the information capacity in the circuit. While it is generally true that the preparation circuit $V(\mathbf{x})$ is unitary, utilizing the parameter shift rule for this purpose is not possible due to the complexity of the circuit's dependence on the input features. In particular, most state preparation circuits will have structures that change based on particular $|x\rangle$ (Buhrman et al., 2024), meaning any differentiation routine will necessarily depend on a complex and changing parameterization. Furthermore, there may be state preparation routines not satisfying the two-eigenvalue criteria mentioned above. In such cases, one would need to use the linear combination of unitaries approach (Schuld et al., 2019), which requires additional matrix decompositions and circuit evaluations. To address this challenge, we provide a novel circuit-based method of calculating the input gradients, which is independent of the routine used for $V(\mathbf{x})$.

**Lemma 3.1** (Input Gradient). *For the general case, assume the amplitudes of an amplitude-encoded input are complex valued, so that each $x_k = c_k + \mathbf{i}\, d_k$. Then, the input gradients with respect to the function given in Eq. 1 are given by the following for the real values and complex-valued components.*

$$\frac{\partial F}{\partial c_k} = 2\operatorname{Re}[\langle b_k| U^\dagger(\boldsymbol{\theta})\, O\, U(\boldsymbol{\theta})\, |x\rangle] \qquad \frac{\partial F}{\partial d_k} = 2\operatorname{Im}[\langle b_k| U^\dagger(\boldsymbol{\theta})\, O\, U(\boldsymbol{\theta})\, |x\rangle]$$

*Proof.* The result is elegant to prove upon judicious use of the product rule for derivatives. A full explicit calculation is deferred to Appendix B. $\square$

Lemma 3.1 gives a compact expression for the $k^{th}$ component of the gradient in terms of the trained model circuit $U(\boldsymbol{\theta})$, its Hermitian conjugate $U^\dagger(\boldsymbol{\theta})$, Hermitian observable $O$, and amplitude embedded state $|x\rangle$. In this work, we are primarily concerned with the case where all amplitudes are real, $x_i = c_i$, as this is the most common case encountered when using classical data.

*Remark* 3.2. If we add the constraint that $O$ be unitary as well as Hermitian, then $U^\dagger(\boldsymbol{\theta})OU(\boldsymbol{\theta})$ corresponds to a valid quantum circuit. The obvious choices for $O$ that satisfy this are Pauli operators or strings of Pauli operators (Dion et al., 2024), which are available on most devices as both measurement and gate operations.

## 4 CALCULATING ON QUANTUM HARDWARE

### 4.1 HADAMARD TEST

**Definition 4.1** (Hadamard Test). *Given unitary operators $A$ and $B$ such that $A|0\rangle = |a\rangle$ and $B|0\rangle = |b\rangle$, the Hadamard test (Montanaro & de Wolf, 2013; Audenaert et al., 2008) is a method for encoding the value $\operatorname{Re}[\langle a|b\rangle]$ into the expectation value of a quantum circuit observable. This is achieved by the following circuit:*

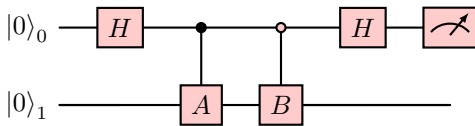

*The circuit for computing $\mathrm{Im}[\langle a|b\rangle]$ is the same, with the addition of an $S^\dagger$ gate after the first $H$ gate (Aharonov et al., 2006).*

After the initial Hadamard gate $H = \frac{1}{\sqrt{2}}\left(\begin{smallmatrix}1 & 1 \\ 1 & -1\end{smallmatrix}\right)$, we have the state $\frac{1}{\sqrt{2}}[|0\rangle_0 + |1\rangle_0]\,|0\rangle_1$. Applying A conditioned on 1 and B conditioned on 0 gives the entangled state

$$\tfrac{1}{\sqrt{2}}[\,|0\rangle_0\,B\,|0\rangle + |1\rangle_0\,A\,|0\rangle\,] = \tfrac{1}{\sqrt{2}}[\,|0\rangle_0\,|b\rangle + |1\rangle_0\,|a\rangle\,].$$

The final Hadamard gate gives us

$$\tfrac{1}{\sqrt{2}}\left(\tfrac{1}{\sqrt{2}}(|0\rangle_0 + |1\rangle_1)\,|b\rangle_1 + \tfrac{1}{\sqrt{2}}(|0\rangle_0 - |1\rangle_1)\,|a\rangle_1\right) = \tfrac{1}{2}(|0\rangle\,(|b\rangle + |a\rangle) + |1\rangle\,(|b\rangle - |a\rangle)).$$

From this we can compute the probability of measuring qubit 0 to be 0 as

$$P(0) = \tfrac{1}{2}[\,\langle b| + \langle a|\,]\cdot\tfrac{1}{2}(|b\rangle + |a\rangle)\,] = \tfrac{1}{4}[\,\langle b|b\rangle + \langle b|a\rangle + \langle a|b\rangle + \langle a|a\rangle\,] = \tfrac{1}{2}[\,1 + \mathrm{Re}[\langle a|b\rangle]\,].$$

This allows us to estimate the desired inner product by sampling from the probability distribution of additional qubits entangled with the system (Schuld et al., 2019). Hadamard tests have been used previously to compute certain kinds of parameter gradients (Bharti et al., 2022; Schuld et al., 2019), but not for feature gradients. We re-frame our formulation in Lemma 3.1 in order to allow for hardware native calculations of the gradients for input amplitudes.

## 4.2 Gradient Calculation for Input Attribution

Lemma 3.1, in conjunction with Definition 4.1, implies that we can calculate the feature gradient of a quantum model using circuit evaluations. We propose a circuit based on a Hadamard test:

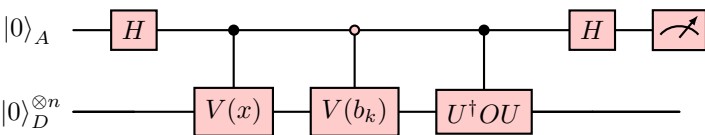

Here, $H$ is the aforementioned Hadamard gate, used to create equal superposition states. $V(x)$ is the preparation circuit that prepares $|x\rangle$. Similarly, $V(b_k)$ prepares the computational basis state $|b_k\rangle$. The wires that extend from one qubit register to another indicate control operations; these are multiqubit operations where the state of one or more qubits in a target register undergoes some transformation, predicated on the state of the control register. In the above circuit, the control is always the ancilla qubit (indexed by $A$), while the targets are always the data qubits (indexed by $D$). The target state that triggers the control operation is indicated by the circle: filled circles indicate control gates triggered by the $|1\rangle_A$ state, while empty circles indicate control gates triggered by $|0\rangle_A$. As an example, consider the first controlled gate, controlled $V(\mathbf{x})$. This gate prepares $|x\rangle$ on the data register $D$ when the ancilla $A\,|1\rangle$, and does nothing when $A$ is in the $|0\rangle$ state.

The key feature of our technique is that gradients are computed from expectation values. While this still requires a controlled version of the quantum model, it is a direct expectation value measurement, which avoids the resource-intensive process of hidden-state tomography or classical simulation. In essence, the difficulty in implementing the technique centers around the controlled operations, which, for a given architecture, might be designed/chosen so that it may be more efficiently compiled or approximated than general multi-controlled gates, especially in hardware that supports native multi-qubit interactions. While it is true that multicontrol gates do incur high overhead on low connectivity platforms like superconducting hardware, it is possible that this overhead might be avoidable on other hardware platforms, like neutral atoms (Delakouras et al., 2025; Levine et al., 2019) and trapped ions (Borrelli et al., 2011; Katz et al., 2023), where native multi-qubit operations are possible. For demonstration purposes, we focus on layers of CNOTs due to their prevalence in the literature, however, it is entirely possible to substitute this for another type of entangling gate, based on these native hardware constraints.

**Theorem 4.2.** *The above circuit returns the $k^{th}$ element of the gradient provided in Lemma 3.1, encoded in the probability of the event where qubit $A$ is measured as 0.*

*Proof.* The result can be seen almost directly from considering definition 4.1. An explicit calculation of the result is provided in Appendix C. The resulting measurement probability on the $A$ register is

$$P(A = 0) = \tfrac{1}{2}(1 + \mathrm{Re}[\,\langle b_k|\,U^\dagger O U\,|x\rangle\,])$$

Comparing with Lemma 3.1, we see that the $k^{th}$ entry of the gradient is contained within the probability of measuring the ancilla to be 0. We can repeat this procedure for each of the components. For a fixed number of measurement shots, this gives a linearly scaling relationship with the number of input features, i.e., one circuit required per input feature, when a single ancilla qubit is used. □

### 4.3 GRADIENT CALCULATION PARALLELIZATION

We can further parallelize the component operations (the $k$'s in Theorem 4.2) by increasing the number of ancilla qubits. Doing so recovers the exponential scaling of input features with (ancilla) qubit count. For instance, if three components need to be executed in parallel, the following circuit is capable of calculating the $k^{th}$, $l^{th}$, and $m^{th}$ components concurrently. The circuit uses two ancilla qubits instead of one and measures both the ancilla simultaneously.

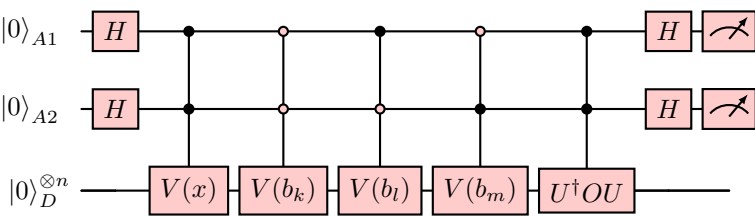

A similar calculation as provided in Theorem 4.2 gives an output probability of

$$P(A_1 A_2 = 00) = \frac{1}{16}\left[4 + 2\,\mathrm{Re}[\langle b_k| \tilde{O} |x\rangle + \langle b_l| \tilde{O} |x\rangle + \langle b_m| \tilde{O} |x\rangle]\right].$$

Similar expressions exist for $P(A_1 A_2 = 01)$, $P(A_1 A_2 = 10)$, and $P(A_1 A_2 = 11)$, from which we can compute the expectation values needed. The number of expectation values extracted from the system is equal to the number of encoded basis states, meaning it scales exponentially with the number of ancilla qubits. For $m$ ancilla qubits, we are able to compute $2^m - 1$ gradient components, preserving the exponential space savings associated with amplitude embedding. The general formula for computing the expectation values is

$$\mathrm{Re}[\langle b_k| \tilde{O} |x\rangle] = 2^{m-1}\sum_a (-1)^{a\cdot(k\oplus r)}\left(p_a - \frac{1}{2^m}\right) \quad \forall\, k \neq r, \tag{4}$$

where $r = 1^m$ is the all ones bitstring, which is reserved to prepare the state $\tilde{O} |x\rangle$. We also have $p_a = P(A_{m-1}A_{m-2}...A_0)$ as the probability of ancilla bitstring $a$. Here the $\oplus$ and $\cdot$ in the exponent denote bitwise XOR and AND respectively. For more detailed calculations, please see Appendix D.

Increasing the number of ancillae will also increase the compilation overhead of the HATTRIQ circuit. This is especially pertinent in the near term, where circuit depth and gate count will limit the length of computations that can be run on hardware. As such, we highlight that the flexibility of HATTRIQ with respect to the number of ancilla qubits used is highly desirable, as it allows users to fine-tune the usage of quantum resources. This aligns with a common design principle of the so-called Early Fault Tolerant Quantum Computing (EFTQC) era, which is to trade reduced circuit complexity and qubit count for increased circuit evaluations (shots) (Liang et al., 2024).

## 5 EVALUATION AND DISCUSSION

### 5.1 EXPERIMENTAL SETUP

We test our technique on a variety of image datasets, including Bars and Stripes (Bowles et al., 2024), MNIST (LeCun, 1998), NIST (similar to MNIST, but with reduced resolution: $8\times8$), and FashionMNIST (Xiao et al., 2017). For each dataset, we construct binary classification tasks by randomly selecting two classes. Training, inference, and gradient calculations are all performed in simulation, assuming ideal error-corrected hardware. As simulation is computationally prohibitive for larger systems, we focus on smaller, but representative, datasets to evaluate HATTRIQ. All simulation code is written in Python 3.12.1, using Qiskit 2.0.0 (Javadi-Abhari et al., 2024) and PennyLane 0.41.1 (Bergholm et al., 2022). Data preprocessing was performed with scikit-learn 1.6.1 (Pedregosa et al., 2011), and the optimization for training circuit parameters was performed

Table 1: Datasets and models used for HATTRIQ's evaluation, including the classification accuracies achieved by the QML classifier models.

| Dataset | Binary Classes | Encoding | Circuit Structure | | Model Accuracy (%) | |
|---------|----------------|----------|---------|----------|----------|---------|
| | | | # Qubits | # Layers | Training | Testing |
| *Bars & Stripes* | (Bars, Stripes) | Amplitude | 4 | 8 | 96 | 95 |
| | | Angle | 8 | 8 | 95 | 95 |
| *NIST* | (0,1), (3,4), (5,6), (6,9), (1,7) | Amplitude | 6 | 6 | 98, 100, 98, 96, 93 | 99, 100, 100, 98, 88 |
| *MNIST* | (0,1), (3,4), (5,6), (6,9), (1,7) | Amplitude | 10 | 10 | 92, 88, 87, 62, 87 | 91, 82, 87, 68, 83 |
| *Fashion MNIST* | (Dress,Shirt), (Boot,Trousers), (Coat,Sandal), (Bag,Sandal), (Boot,Dress) | Amplitude | 10 | 10 | 74, 100, 96, 74, 90 | 70, 99, 95, 69, 91 |

using COBYLA (COB, 1994), as implemented in Scipy 1.15.1 (Virtanen et al., 2020). Due to the difficulty of training the angle-embedded model, we used a gradient descent optimizer implemented in PennyLane. Experiments are run on a local cluster, consisting of nodes with the AMD EPYC 7702P 64-core processor. We spawn virtual machines with 8 cores and 32 GB of memory.

## 5.2 MODEL ARCHITECTURE

To focus on the general applicability of our technique, we choose to train relatively simple models (model properties are shown in Table 1) based on the hardware-efficient ansatz, which is composed of alternating layers of single-qubit rotation gates and two-qubit CNOT gates (Arrasmith et al., 2021). An example of this structure is shown in Appendix E for reference. Data is encoded into the system with an amplitude encoding scheme, where the intensity of a pixel corresponds to the amplitude of one of the basis states. For these datasets, it is not generally true that each data point is normalized with $|\mathbf{x}| = 1$, meaning we can not directly encode them as quantum states $|x\rangle = \sum_i x_i |b_i\rangle$, but must first apply some transformation. The easiest of these is to simply divide each data point by its norm; however, we find empirically that this can cause issues during training, as the absolute value of a pixel's amplitude can change from image to image, even when the intensity is the same, due to images having differing levels of overall brightness. We instead utilize an encoding scheme that has an overflow state. This overflow state allows us to encode the value of each pixel in a way that is consistent image to image, while maintaining the normalization condition of quantum states. In an $n$ qubit model, we scale $2^n - 1$ pixel values $x_i$ to be within $[0, (\frac{1}{2^n-1})^{\frac{1}{2}}]$. The remaining state, the overflow state, is then assigned the value $(1 - \sum_i^{2^n-1} |x_i|^2)^{\frac{1}{2}}$ so that the final norm of the state is 1. Measurement is performed on a single qubit in the Z basis, i.e., $O$ in Eq. 1 is the single qubit Z operator. While testing, we found improved performance when using a nonlinear tanh activation applied to the output of the circuit. All of our discussion from before still applies upon simple modification using the chain rule.

## 5.3 HATTRIQ'S ATTRIBUTION RESULTS

We show attribution scores for a variety of samples from each of the datasets. These samples are chosen randomly for analysis. We use a blank image (0 for all pixel values) for the baseline in all tests. While developing our technique, we also experimented with alternative baselines, including the average of all images in the training set. Results for this alternative baseline are excluded here for brevity, but are presented in Appendix G.

Positive and negative attributions refer to an input's contribution toward the model's prediction, just as they do in the deep learning setting. Our model output is $F(x; \theta) = \langle x|U^\dagger(\theta)OU(\theta)|x\rangle$. This model prediction also corresponds to the expected measurement outcome of the physical system, as

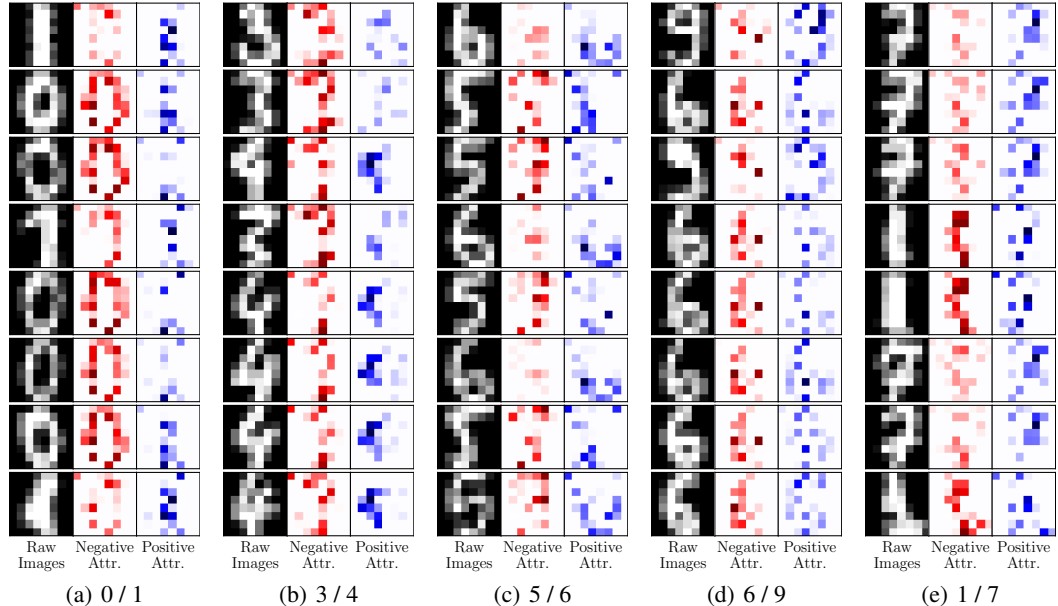

|  |  |  |  |  |
|---|---|---|---|---|
| Raw Images | Negative Attr. | Positive Attr. | Raw Images | Negative Attr. | Positive Attr. | Raw Images | Negative Attr. | Positive Attr. | Raw Images | Negative Attr. | Positive Attr. | Raw Images | Negative Attr. | Positive Attr. |
| (a) 0 / 1 | (b) 3 / 4 | (c) 5 / 6 | (d) 6 / 9 | (e) 1 / 7 |

Figure 1: Sample images and the accompanying integrated gradients attribution for various samples from the NIST dataset. Quantum models were trained for various binary classification tasks. Blue indicates positive attribution, red indicates negative attribution, and white indicates neutral attribution. We see patches and patterns of strong attributions for the trained classifier models.

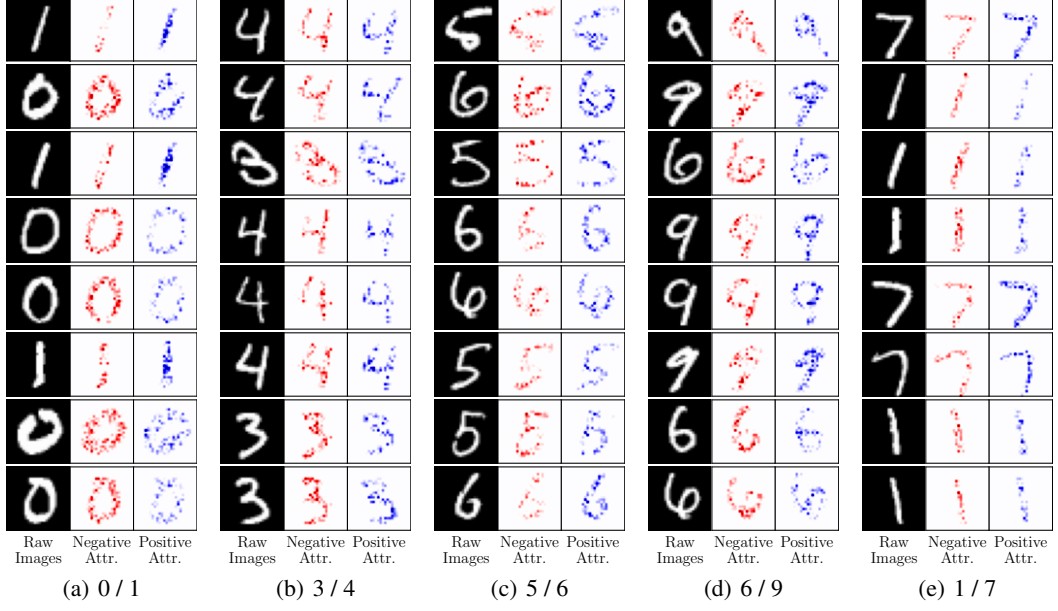

|  |  |  |  |  |
|---|---|---|---|---|
| Raw Images | Negative Attr. | Positive Attr. | Raw Images | Negative Attr. | Positive Attr. | Raw Images | Negative Attr. | Positive Attr. | Raw Images | Negative Attr. | Positive Attr. | Raw Images | Negative Attr. | Positive Attr. |
| (a) 0 / 1 | (b) 3 / 4 | (c) 5 / 6 | (d) 6 / 9 | (e) 1 / 7 |

Figure 2: Sample images and gradient attribution for the amplitude-embedded MNIST dataset.

this result is what is used as the prediction value. Thus, we attribute how redistributing amplitude (and hence probability mass via the Born rule) onto basis configuration $|b_k\rangle$ changes the observable $O$ along the input path. Positive IG means increasing the feature $x_k$ increases the class score; negative IG means the opposite.

In all plots, negative attributions are plotted in red, while positive attributions are plotted in blue. For visual clarity, attributions are normalized within each sample. Fig. 1 shows the integrated gradient outputs for a variety of samples from the NIST dataset. We see that background pixels have very little importance, as we might expect. We also see that the model has identified features that correspond to the target classes; an example being Fig. 1(b), where we see negative attributions corresponding to

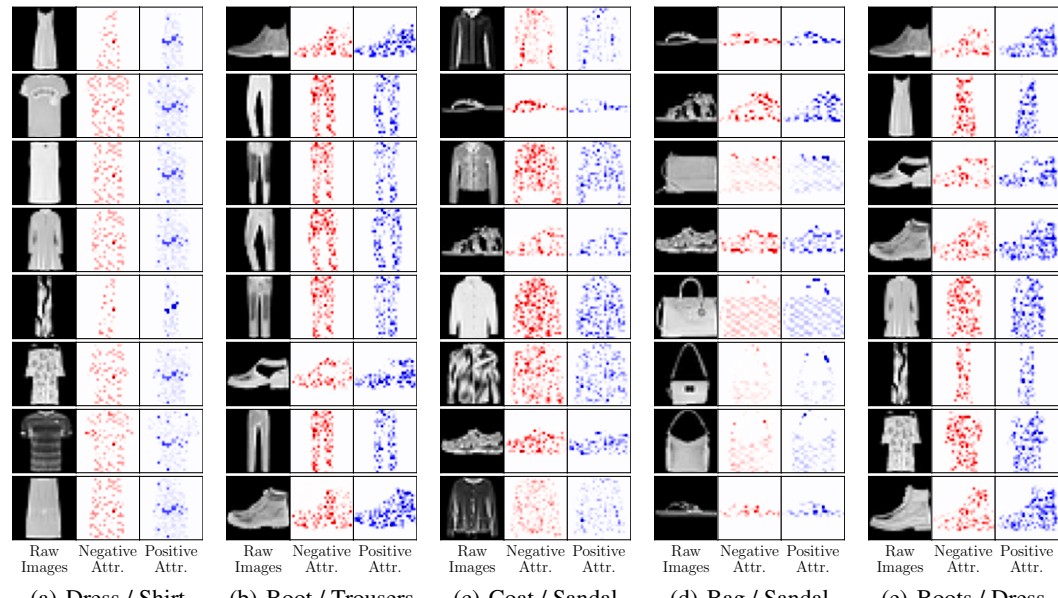

(a) Dress / Shirt  (b) Boot / Trousers  (c) Coat / Sandal  (d) Bag / Sandal  (e) Boots / Dress

Figure 3: Sample images and attributions for the FashionMNIST dataset using amplitude encoding.

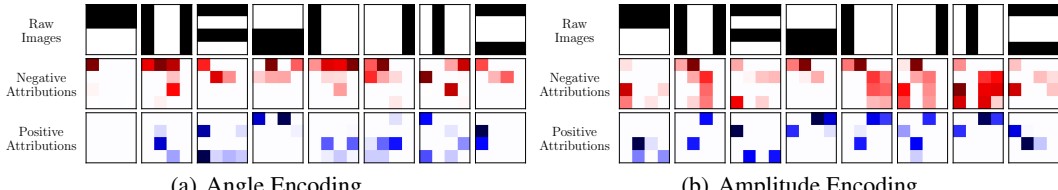

(a) Angle Encoding  (b) Amplitude Encoding

Figure 4: Sample images and the accompanying integrated gradients attribution for the Bars and Stripes dataset. Quantum models using (a) angle encoding and (b) amplitude encoding were trained.

the circular shape of the digit 3 toward the upper right, and positive attributions corresponding to pixels near the center left, corresponding to the angled shape of the digit 4.

We see similar trends with MNIST (Fig. 2) and FashionMNIST (Fig. 3), with the model attributing regions of each image to each class. In these larger examples, the attributions appear more mixed spatially. This is especially noticeable in the Dress/Shirt task, which shows a banding pattern forming, in addition to clusters of strong attributions at the center. We observe that in some cases, the model identifies distinctive features. One example is the Bag/Sandal task, where we see high attribution along the straps, which are only ever present on bags, but never on sandals. We also observe this in the Coat/Sandal task, where the upper area consistently receives negative attributions; this area is unlikely to contain any part of a sandal due to its low profile near that end of the shoe.

We compare the attribution scores of a model using angle embedding and a model using amplitude embedding to see if there are differences in attributions created by different encoding schemes. Despite achieving very similar final accuracy scores (Table 1), we see markedly different attributions for the Bars and Stripes dataset in Fig. 4.

## 5.4 THE EFFECT OF NOISE

To quantify the resource usage of HATTRIQ, we study the impact of using reduced measurement shots on the ancilla qubit. We compute attribution scores using 10, 100, and 500 shots to estimate each component $\text{Re}[\langle b_k | U^\dagger O U | x \rangle]$. Just as before, we repeat this for each component $k$ of the gradient, and use numerical integration to compute the IG attribution. We compare against exact simulation, which numerically computes all inner products directly from the state vector. The results of this are shown in Fig. 5. We observe that even with an extremely low shot count, the attribution scores

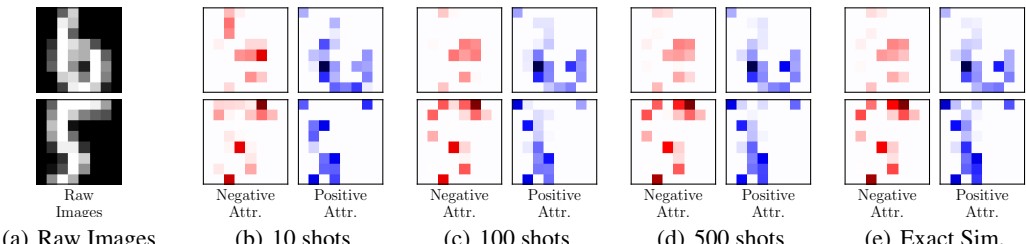

(a) Raw Images     (b) 10 shots     (c) 100 shots     (d) 500 shots     (e) Exact Sim.

Figure 5: Integrated gradients computed using various amounts of measurement shots (samples). In (b), (c), and (d), gradient components are computed using our circuit-based approach, using 10, 100, and 500 samples to estimate each component. Overall, we observe minimal degradation in the attribution scores compared to those obtained by exact simulation (e).

computed are largely faithful to the numerically exact ones, with only small deviations appearing in some of the weaker attributions (those with a lower absolute value).

To further validate HATTRIQ's attribution scores, we also compute attributions for a null model having randomly generated parameters. These results are provided in Appendix F.

We also study the impact of hardware noise on HATTRIQ's performance, by simulating HATTRIQ's IG calculation with the addition of a depolarizing noise model. We find that with modest error rates ($10^{-4}$), attributions remain faithful to the ideal calculation. These results and a description of the error model are provided in Appendix H.

## 6   RELATED WORK

Several recent works have explored interpretability in QML, though none target input attribution directly, and none provide a general, hardware-compatible gradient-based solution as in HATTRIQ. Recent efforts in QML interpretability span model-agnostic techniques, gradient-based methods, and visualization tools. Pira et al. (Pira & Ferrie, 2024) and Jahin et al. (Jahin et al., 2023) apply classical attribution methods like LIME and SHAP to QML models, while Heese et al. (Heese et al., 2025) use Shapley values to explain circuit components. These approaches rely on perturbation-based estimates and rely on surrogate-based analysis, hence are not designed for execution on quantum hardware.

Gradient-based methods, such as QGrad-CAM (Lin et al., 2024), demonstrate attribution in hybrid models using class activation maps, but are limited to specific architectures and do not generalize to amplitude encoding schemes. Visualization-driven efforts like QuantumEyes (Ruan et al., 2023) and interpretable model designs (Flamini et al., 2024; Duneau et al., 2024; Flam-Shepherd et al., 2022; Ran & Su, 2023) focus on circuit behavior or latent representations rather than input-level attribution and are limited in hardware compatibility (e.g., photonics or trapped ions). *In contrast,* HATTRIQ *provides the first gradient-based input attribution method for QML models.* It supports amplitude encoding and enables scalable attribution via Hadamard test circuits and parallel gradient evaluation, making it broadly applicable across quantum models and devices.

## 7   CONCLUSION

We presented HATTRIQ, a unified framework for gradient-based feature attribution in quantum machine learning models. As the first-of-its-kind quantum interpretability method, HATTRIQ operates on exponentially scaling amplitude encoding schemes and is designed for execution on quantum hardware, offering circuit-based gradient computations. We plan to extend HATTRIQ to generate parameter/layer attributions for QML models to determine their importance on the QML task with potential disagreements between attributions from multiple runs (Krishna et al., 2022). We also plan to extend HATTRIQ to support QML models with mid-circuit measurements and conditional gate operations, which are starting to become available on quantum hardware. Due to the effectiveness and unitary nature of QML, it is also of interest to explore unitary feature learning, or equivalently, learning from spherical features (Luo et al., 2024). By leveraging a Hadamard test–based construction and a multi-ancilla parallelization strategy, HATTRIQ enables scalable, implementation-agnostic input attribution with fidelity guarantees.

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

## A FURTHER BACKGROUND ON QUANTUM COMPUTING

Quantum computing is a computational paradigm that leverages the principles of quantum mechanics to process information. The fundamental unit of information is the *quantum bit* (qubit), which can exist in a superposition of basis states,

$$|\psi\rangle = \alpha|0\rangle + \beta|1\rangle, \qquad |\alpha|^2 + |\beta|^2 = 1.$$

Here, $|0\rangle$ and $|1\rangle$ are basis states. They are typically represented with matrices:

$$|0\rangle = \begin{bmatrix} 1 \\ 0 \end{bmatrix}, \quad |1\rangle = \begin{bmatrix} 0 \\ 1 \end{bmatrix}.$$

To describe systems of multiple qubits, we use a *tensor product*. If two qubits are in states $|\psi\rangle$ and $|\phi\rangle$, the joint state is

$$|\psi\rangle \otimes |\phi\rangle,$$

which spans a $2^2$–dimensional Hilbert space. Extending to $n$ qubits yields a $2^n$–dimensional space. Quantum computation proceeds through *unitary gates*, reversible operations that evolve quantum states. A fundamental example is the Hadamard gate,

$$H = \frac{1}{\sqrt{2}} \begin{pmatrix} 1 & 1 \\ 1 & -1 \end{pmatrix},$$

which creates a superposition from $|0\rangle$:

$$H|0\rangle = \frac{1}{\sqrt{2}}(|0\rangle + |1\rangle).$$

The circuit diagram for this operation is:

$$|0\rangle - \boxed{H} -$$

Gates can also act on more than one qubit. An example is the Control-NOT (CNOT), used to generate entanglement. Starting from $|00\rangle$, a Hadamard on the first qubit followed by a CNOT on the second produces:

$$|00\rangle \xrightarrow{H \otimes I} \frac{1}{\sqrt{2}}(|00\rangle + |10\rangle) \xrightarrow{\text{CNOT}} \frac{1}{\sqrt{2}}(|00\rangle + |11\rangle).$$

The corresponding circuit is:

The state produced by this circuit demonstrates the key property of *entanglement*. Quantum computation concludes with *measurement*, which converts the quantum state into classical outcomes. Measuring $|\psi\rangle$ in the computational basis yields outcome $b_k$ with probability

$$P(b_k) = |\langle b_k|\psi\rangle|^2.$$

Here, $b_k$ is the $k$th bitstring of length $n$, and $|b_k\rangle$ the associated basis state.$b_k$ Due to this probabilistic nature, many algorithms rely on repeated circuit executions to estimate expectation values or decision outcomes.

## B   PROOF OF LEMMA 3.1: INPUT GRADIENTS OF QUANTUM MODELS

For compactness, define $\tilde{O} = U^\dagger(\boldsymbol{\theta}) \, O \, U(\boldsymbol{\theta})$ Then, after rewriting Eq. 1, we have:

$$
F(\mathbf{x}\,;\boldsymbol{\theta}) = \left( \sum_{i=0}^{2^n-1} \langle b_i | \, x_i^* \right) \tilde{O} \left( \sum_{j=0}^{2^n-1} x_j \, | b_j \rangle \right) = \left( \sum_{i=0}^{2^n-1} \langle b_i | \, (c_i - \mathbf{i}\, d_i) \right) \tilde{O} \left( \sum_{j=0}^{2^n-1} (c_j + \mathbf{i}\, d_j) \, | b_j \rangle \right)
$$

$$
= \textstyle\sum_{i,j} (c_i - \mathbf{i}\, d_i)(c_j + \mathbf{i}\, d_j) \, \langle b_i | \, \tilde{O} \, | b_j \rangle
$$

Taking the derivative with respect to $c_k$:

$$
\frac{\partial F}{\partial c_k} = \textstyle\sum_{ij} \frac{\partial c_i}{\partial c_k} (c_j + \mathbf{i}\, d_j) \, \langle b_i | \, \tilde{O} \, | b_j \rangle + (c_i - \mathbf{i}\, d_i) \frac{\partial c_j}{\partial c_k} \, \langle b_i | \, \tilde{O} \, | b_j \rangle
$$

$$
= \textstyle\sum_{ij} \delta_{ik} (c_j + \mathbf{i}\, d_j) \, \langle b_i | \, \tilde{O} \, | b_j \rangle + \delta_{jk} (c_i - \mathbf{i}\, d_i) \, \langle b_i | \, \tilde{O} \, | b_j \rangle
$$

$$
= \textstyle\sum_j (c_j + \mathbf{i}\, d_j) \, \langle b_k | \, \tilde{O} \, | b_j \rangle + \textstyle\sum_i (c_i - \mathbf{i}\, d_i) \, \langle b_i | \, \tilde{O} \, | b_k \rangle
$$

$$
= \textstyle\sum_j (c_j + \mathbf{i}\, d_j) \, \langle b_k | \, \tilde{O} \, | b_j \rangle + (c_j - \mathbf{i}\, d_j) \, \langle b_j | \, \tilde{O} \, | b_k \rangle
$$

$$
= \textstyle\sum_j 2\,\mathrm{Re}[(c_j + \mathbf{i}\, d_j) \, \langle b_k | \, \tilde{O} \, | b_j \rangle] = 2\,\mathrm{Re}[\langle b_k | \, \tilde{O} \textstyle\sum_j (c_j + \mathbf{i}\, d_j) \, | b_j \rangle]
$$

$$
= 2\,\mathrm{Re}[\langle b_k | \, \tilde{O} \, | x \rangle] = 2\,\mathrm{Re}[\, \langle b_k | \, U^\dagger(\boldsymbol{\theta}) \, O \, U(\boldsymbol{\theta}) \, | x \rangle \,]
$$

Here, $\delta_{ik}$ is the Kronecker delta, and we have made use of the fact that $\tilde{O}^\dagger = \tilde{O}$. A similar derivation exists for $\frac{\partial F}{\partial d_k}$. We exclude it here for brevity.

## C   PROOF OF THEOREM 4.2: HADAMARD TEST COMPUTATION

We use the subscript $A$ for the state of ancilla qubit(s), and the subscript $D$ for the state of data qubit(s). The final state of the circuit from section 4.2 before measurement is given by:

$$
|\psi\rangle = (I \otimes H) \cdot C\tilde{O} \cdot \bar{C}V(b_k) \cdot CV(x) \cdot (I \otimes H) \cdot (|0\rangle_D^{\otimes n} \otimes |0\rangle_A)
$$

$$
= (I \otimes H) \cdot C\tilde{O} \cdot \bar{C}V(b_k) \cdot CV(x) \cdot \tfrac{1}{\sqrt{2}} \left( |0\rangle_D^{\otimes n} \otimes |0\rangle_A + |0\rangle_D^{\otimes n} \otimes |1\rangle_A \right)
$$

$$
= (I \otimes H) \cdot \tfrac{1}{\sqrt{2}} \left( |b_k\rangle_D \otimes |0\rangle_A + \tilde{O} \, |x\rangle_D \otimes |1\rangle_A \right)
$$

$$
= \tfrac{1}{2} \left( |b_k\rangle_D \otimes (|0\rangle_A + |1\rangle_A) + \tilde{O} \, |x\rangle_D \otimes (|0\rangle_A - |1\rangle_A) \right)
$$

$$
= \tfrac{1}{2} \left( (|b_k\rangle + \tilde{O} \, |x\rangle)_D \otimes |0\rangle_A + (|b_k\rangle - \tilde{O} \, |x\rangle)_D \otimes |1\rangle_A \right)
$$

Here, $C$ denotes the control operations that trigger on $|1\rangle_A$ and $\bar{C}$ denotes the control operations that trigger on $|0\rangle_A$. From here, using the standard probability rule, we see that

$$
P(A = 0) = |\tfrac{1}{2}(|b_k\rangle + U^\dagger O U \, |x\rangle)|^2 = \tfrac{1}{4} | \langle b_k | b_k \rangle + \langle x | x \rangle + \langle b_k | \, U^\dagger O U \, |x\rangle + \langle x | \, U^\dagger O U \, |b_k\rangle \,|
$$

$$
= \tfrac{1}{2}(1 + \mathrm{Re}[\, \langle b_k | \, U^\dagger O U \, |x\rangle \,])
$$

## D   FULL DERIVATION OF GRADIENT PARALLELIZATION

Parallel computation of the gradient entries is made possible by increasing the number of ancilla qubits. For the 2 ancilla case, the circuit looks like:

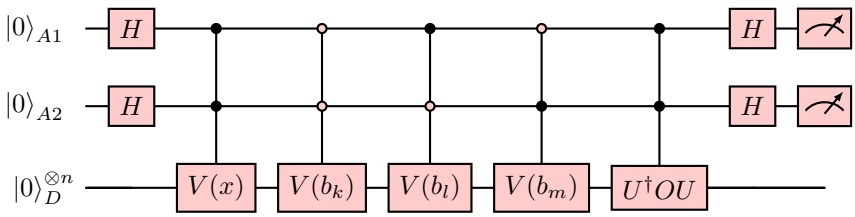

This generalizes to higher ancilla counts by adding additional controls. To formalize this, we state the following theorem:

**Theorem D.1** (Parallel Hadamard-test with $m$ ancillas). *Let $D$ be an $n$-qubit data register and $A$ an $m$-qubit ancilla register. Fix a* reserved *ancilla string $r := 1^m$. For every $s \in \{0,1\}^m \setminus \{r\}$, let $|b_s\rangle$ be an orthonormal computational-basis state for $D$. Let $|x\rangle$ be an arbitrary normalized data state and let $\tilde{O}$ be unitary. Define*

$$r_s := \mathrm{Re}[\langle b_s| \tilde{O} |x\rangle] \quad (s \neq r).$$

*Then a single circuit using $m$ ancillas yields all $2^m - 1$ values $r_s$ (and hence all gradients $g_s = 2r_s$) by measuring $A$.*

*Proof.* Start in $|0^m\rangle_A |0^n\rangle_D$ and apply $H^{\otimes m}$ on $A$:

$$|\Psi_0\rangle = \frac{1}{\sqrt{2^m}} \sum_{s \in \{0,1\}^m} |s\rangle_A |0^n\rangle_D.$$

Apply controlled state-preparations on $D$: if $s \neq r$ prepare $|b_s\rangle$ from $|0^n\rangle$, and if $s = r$ prepare $|x\rangle$. This gives

$$|\Psi_1\rangle = \frac{1}{\sqrt{2^m}} \Big( \sum_{s \neq r} |s\rangle |b_s\rangle + |r\rangle |x\rangle \Big).$$

Apply a controlled-$\tilde{O}$ only on branch $r$:

$$|\Psi_2\rangle = \frac{1}{\sqrt{2^m}} \Big( \sum_{s \neq r} |s\rangle |b_s\rangle + |r\rangle \tilde{O} |x\rangle \Big).$$

Finally apply $H^{\otimes m}$ on $A$. Using $H^{\otimes m} |s\rangle = 2^{-m/2} \sum_a (-1)^{a \cdot s} |a\rangle$ (the inverse Hadamard transform), we obtain

$$|\Psi_3\rangle = \frac{1}{2^m} \sum_a |a\rangle \Big( \sum_{s \neq r} (-1)^{a \cdot s} |b_s\rangle + (-1)^{a \cdot r} \tilde{O} |x\rangle \Big).$$

Define the (unnormalized) conditional data state

$$|v_a\rangle := \sum_{s \neq r} (-1)^{a \cdot s} |b_s\rangle + (-1)^{a \cdot r} \tilde{O} |x\rangle.$$

Then the ancilla outcome probability is

$$p_a = \mathrm{Pr}(A = a) = \frac{\| |v_a\rangle \|^2}{2^{2m}}.$$

Because the $\{|b_s\rangle\}_{s \neq r}$ are orthonormal and $\|\tilde{O} |x\rangle \| = 1$ (since $\tilde{O}$ is unitary), we have

$$\|v_a\|^2 = \sum_{s \neq r} \|b_s\|^2 + \|\tilde{O}x\|^2 + 2 \, \mathrm{Re}[\sum_{s \neq r} (-1)^{a \cdot s} (-1)^{a \cdot r} \langle b_s| \tilde{O} |x\rangle]$$

$$= (2^m - 1) + 1 + 2 \sum_{s \neq r} (-1)^{a \cdot (s \oplus r)} r_s$$

$$= 2^m + 2 \sum_{s \neq r} (-1)^{a \cdot (s \oplus r)} r_s.$$

Hence

$$p_a = \frac{1}{2^m} + \frac{1}{2^{2m-1}} \sum_{s \neq r} (-1)^{a \cdot (s \oplus r)} r_s \tag{5}$$

We can invert Eq. 5 as follows. Let $\Delta_a := p_a - 2^{-m}$. Multiply Eq. 5 by $(-1)^{a \cdot (t \oplus r)}$ and sum over all $a$:

$$\sum_a (-1)^{a \cdot (t \oplus r)} \Delta_a = \frac{1}{2^{2m-1}} \sum_{s \neq r} r_s \sum_a (-1)^{a \cdot (s \oplus t)}.$$

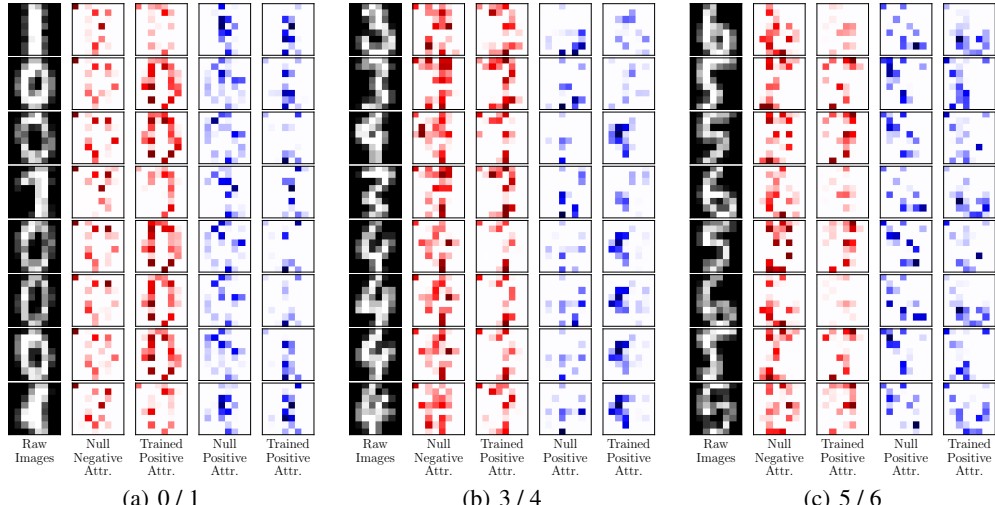

| Raw Images | Null Negative Attr. | Trained Positive Attr. | Null Positive Attr. | Trained Positive Attr. | Raw Images | Null Negative Attr. | Trained Positive Attr. | Null Positive Attr. | Trained Positive Attr. | Raw Images | Null Negative Attr. | Trained Positive Attr. | Null Positive Attr. | Trained Positive Attr. |

(a) 0 / 1        (b) 3 / 4        (c) 5 / 6

Figure 6: Sample images and the accompanying integrated gradients attribution for various samples from the NIST dataset. Attributions are for untrained null models with parameters sampled from a uniform distribution on the interval $[0, \pi)$. For comparison, we re-plot attributions for trained models from Fig. 1 alongside the null model attributions. Blue indicates positive attribution, red indicates negative attribution, and white indicates neutral attribution. We see from the lack of concentration that the null models fail to identify key features.

Using the identity $\sum_a (-1)^{a \cdot (s \oplus t)} = 2^m \delta_{s,t}$, we find that, for any $t \neq r$

$$r_t = 2^{m-1} \sum_a (-1)^{a \cdot (t \oplus r)} \left( p_a - \frac{1}{2^m} \right). \tag{6}$$

Thus measuring all ancillas provides the full probability vector $(p_a)$, from which classical post-processing generates every $r_t$ simultaneously. Since the attribution gradients satisfy $g_t = 2r_t$ (from Lemma 3.1), this yields $2^m - 1$ gradient components in parallel using only $m$ ancillas. $\square$

# E    CIRCUIT STRUCTURE USED FOR QML MODELS

Our trained circuits are all constructed from a hardware-efficient ansatz, which consists of alternating rows of single-qubit rotations and two-qubit CNOT gates. These layers are repeated multiple times to increase the number of model parameters. An example with 6 qubits is shown below:

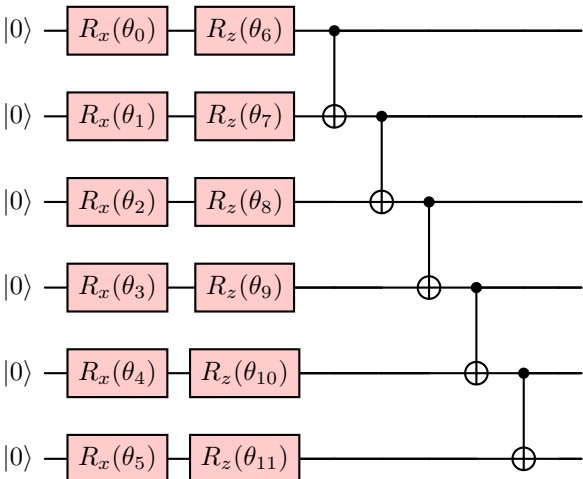

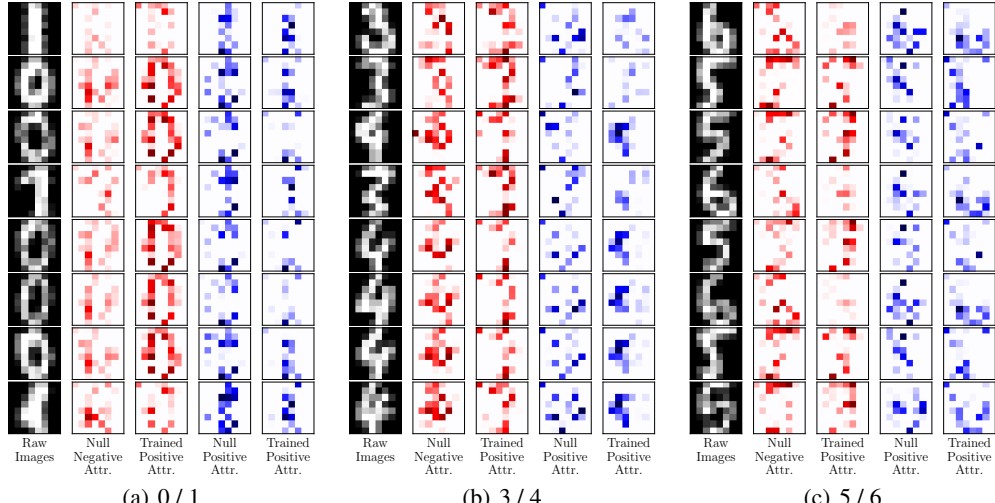

(a) 0 / 1          (b) 3 / 4          (c) 5 / 6

Figure 7: Sample images and the accompanying integrated gradients attribution for various samples from the NIST dataset. Attributions are for untrained null models with parameters sampled from a Gaussian distribution $\mathcal{N}(0, \frac{\pi}{2}^2)$. For comparison, we re-plot attributions for trained models from Fig. 1 alongside the null model attributions. Blue indicates positive attribution, red indicates negative attribution, and white indicates neutral attribution. We see from the lack of concentration that the null models fail to identify key features.

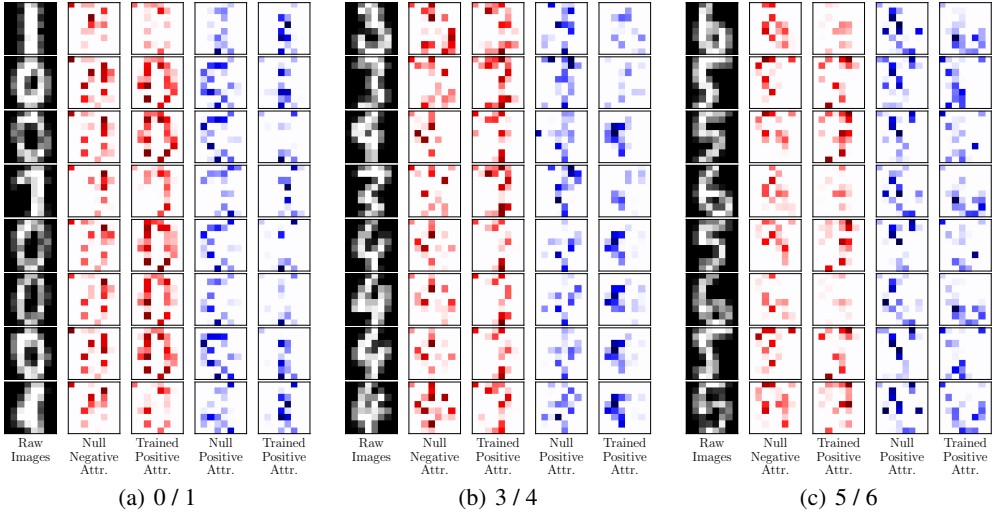

(a) 0 / 1          (b) 3 / 4          (c) 5 / 6

Figure 8: Sample images and the accompanying integrated gradients attribution for various samples from the NIST dataset. Attributions are for untrained null models with parameters sampled from Student's distribution with $\nu = 2$. For comparison, we re-plot attributions for trained models from Fig. 1 alongside the null model attributions. Blue indicates positive attribution, red indicates negative attribution, and white indicates neutral attribution. We see from the lack of concentration that the null models fail to identify key features.

In such a circuit, the number of parameters is proportional to the number of qubits × the number of layers. Generally, selecting a layer count between 1× and 2× times the number of qubits provides the best accuracy (as demonstrated by our selection for the number of layers in Table 1).

## F  VALIDATION AGAINST NULL MODEL ATTRIBUTIONS

To validate HATTRIQ's attribution scores, we also compute attributions for null models having randomly generated parameters, shown in Fig. 6. Parameters are randomly sampled from either a uniform distribution on the interval $[0, \pi)$, a normal distribution $\mathcal{N}(0, \frac{\pi}{2}^2)$, and a heavy-tailed Student's t-distribution ($\nu = 2$), Attributions are then computed and plotted for the same samples as used in Fig. 1, as shown in Figs. 6, 7, and 8 for the uniform, normal, and t-distributions respectively. Across the various classification tasks, we fail to see notable concentration or clustering of the attribution scores with either of the three distributions, unlike the trained case with HATTRIQ. For instance, the angular edge on the left side of digit four is only identified and attributed by HATTRIQs, while the three null attributions provide attribution scores all across the image.

## G  BASELINE SELECTION

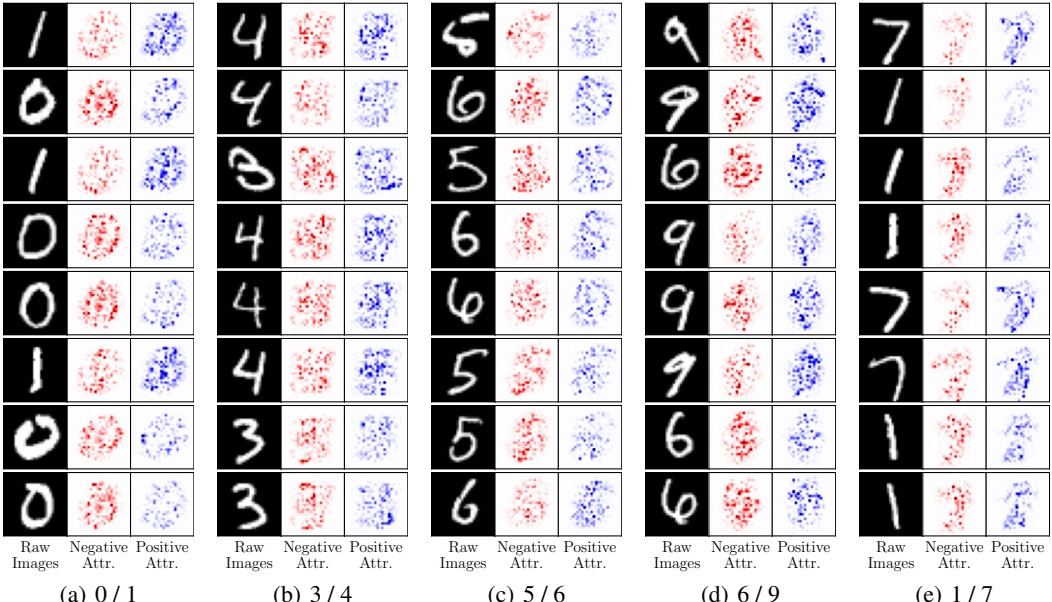

| Raw Images | Negative Attr. | Positive Attr. | Raw Images | Negative Attr. | Positive Attr. | Raw Images | Negative Attr. | Positive Attr. | Raw Images | Negative Attr. | Positive Attr. | Raw Images | Negative Attr. | Positive Attr. |

(a) 0 / 1          (b) 3 / 4          (c) 5 / 6          (d) 6 / 9          (e) 1 / 7

Figure 9: Sample images and gradient attribution for the amplitude-embedded MNIST dataset, using an average of the training images as a baseline. We have merged the attributions to show positive and negative attributions in the same image.

Integrated gradients are computed relative to a baseline image. For the examples shown in the main paper, we utilize a blank image baseline, where all pixels are set to 0. Here, we provide additional results that utilize an average baseline, where the pixels are set to the average value calculated over all the training images used in the set. We show these results for MNIST in Fig. 9.

## H  NOISY SIMULATION

To probe the robustness of our technique in noisy operating conditions, we additionally simulate HATTRIQ's performance under hardware noise representative of the early fault-tolerant quantum computing era. In this regime, logical operations remain subject to non-negligible residual error rates despite the presence of error correction, reflecting the expected constraints of near-term fault-tolerant devices where overheads preclude arbitrarily deep suppression (Liang et al., 2024). We model these effects using a standard depolarization channel, as implemented in Qiskit Aer (Javadi-Abhari et al., 2024):

$$\mathcal{E}[\rho] = (1 - \gamma)\rho + \gamma \operatorname{Tr}[\rho] \frac{I}{2^n}, \tag{7}$$

where $\rho$ is the density matrix of the system, $\gamma$ is the error rate and $n$ is the number of qubits. We focus on a regime of mild noise ($\gamma = 10^{-4}$) to simulate devices less noisy than the current generation,

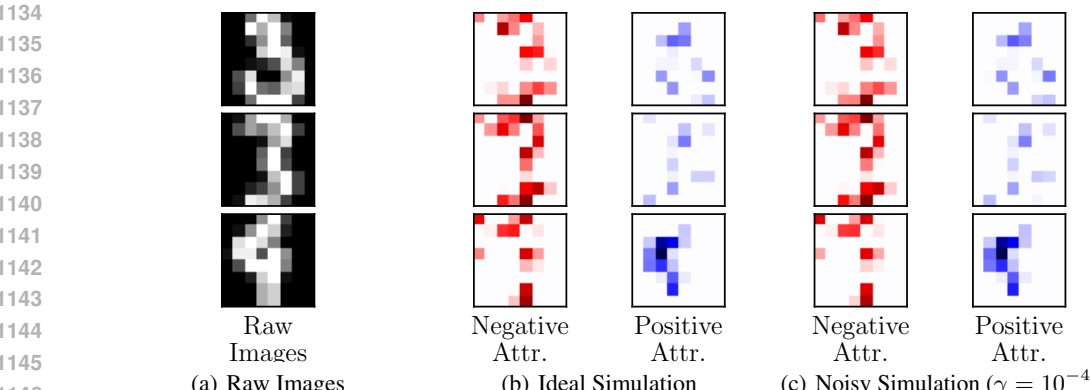

|       |       |       |       |       |
|-------|-------|-------|-------|-------|
| Raw Images | Negative Attr. | Positive Attr. | Negative Attr. | Positive Attr. |
| (a) Raw Images | (b) Ideal Simulation | | (c) Noisy Simulation ($\gamma = 10^{-4}$) | |

Figure 10: Integrated gradients computed in for three samples (a) of the NIST 3/4 classification task. In (b) simulation is performed assuming ideal hardware. In (c), a depolarizing error channel is added to the simulation, with error rate $10^{-4}$. Overall, we observe minimal degradation in the noisy hardware attribution scores compared to the ideal case.

but not yet fully fault-tolerant. To keep our discussion general, we transpile all circuits to the basis $\{U, CNOT, CCX\}$, and assume full qubit connectivity. This choice of basis also makes transpiling our circuit straightforward, as we can simply attach an additional control from the ancilla to each gate in the ansatz (shown in Appendix E). The controlled rotations are then transpiled further, while the CCX gates remain. We demonstrate integrated gradient calculations on the NIST 3/4 classification task, using a single ancilla for computation. Notably, the resulting performance, shown in Fig. 10, closely aligns with that observed in idealized, noiseless simulations: the generated scores deviate only mildly, preserving spatial distribution and relative intensity. These findings indicate that our method is well-suited to early fault-tolerant noise levels.

## I  LLM USAGE

We would like to mention that an LLM, specifically Grammarly AI, was used to aid and polish writing, specifically to correct grammatical mistakes. We also utilized OpenAI GPT-5.1 Thinking when formalizing the mathematical derivation of the parallel gradients in Appendix D. All mathematical steps were verified by the authors.

## J  REPRODUCIBILITY STATEMENT

All code and datasets are open-sourced and included with this work, with instructions and scripts for reproducing results. This ensures transparency, accelerates future research, and enables broad community adoption.

