# OpenReview forum: "Amplitude-based Input Attribution in Quantum Learning via Integrated Gradients"
_ICLR.cc/2026/Conference — Submitted to ICLR 2026_

### Official Review · Reviewer_Nf8r · 2025-10-23

**Soundness:** 3
**Presentation:** 3
**Contribution:** 2
**Rating:** 4
**Confidence:** 4

**Summary:**

This paper introduces HattriQ, a novel framework for computing input attribution scores in QML models. HattriQ adapts the classical Integrated Gradients method to the quantum setting, specifically for models using amplitude embedding, an encoding scheme that allows for an exponential number of input features relative to the number of qubits. The core of the method is a quantum-native circuit construction based on the Hadamard test to compute the exact gradients of the model output with respect to the input features directly on quantum hardware, without requiring access to or simulation of the internal quantum state. The method is evaluated across datasets such as bars and stripes, MNIST and FashionMNIST.

**Strengths:**

1. This work proposes to use quantum-native Hadamard tests to evaluate gradients of the cost function w.r.t. input features. It primarily targets amplitude encoding, but also generalizes angle encoding strategies.

2. To enhance scalability, the authors propose a parallelization strategy utilizing multiple ancilla qubits, which mitigates the linear scaling of circuit evaluations by enabling the concurrent computation of numerous gradient components

3. Comprehensive experiments are performed on diverse benchmark datasets.

**Weaknesses:**

A significant weakness of the proposed method lies in its fundamental scalability, which stem directly from its targeting of amplitude encoding. While this encoding allows for an exponential data capacity in principle, the attribution process requires computing a gradient component for each of the exponentially many amplitude features. Although the multi-ancilla parallelization strategy offers a theoretical mitigation, it necessitates a number of ancilla qubits that scales linearly with the number of features to be computed concurrently, presenting a substantial resource demand.

**Questions:**

No.

**Details Of Ethics Concerns:**

No.

---

> ### Author Response · Authors · 2025-11-21
>
> Dear Reviewer Nf8r,
>
> We are grateful for your comments and insights. Below, we outline the revisions we implemented and provide clarifications on the scalability of our technique
>
>
> **Ancillae Requirements**
>
> We would like to emphasize that the number of features attributed scales exponentially with the number of ancillae, not linearly, so that the space savings of the quantum model are not negated. To see this, note that in the circuit diagram of section 4.3 each ancilla bitstring corresponds to one prepared basis state, with the exception of the “11” string, which is used to prepare the data. The number of features extracted scales with the number of ancilla basis states; that is exponentially with the number of ancilla qubits. We have revised part of section 4.3 to emphasize this point, and provided a more formal derivation in Appendix D.
>
> A common theme of Early Fault Tolerant Quantum Computing is to [trade reduced depth/error correction overhead for increased sampling complexity/runtime ](https://doi.org/10.1103/PhysRevResearch.6.023118). HattriQ provides a natural and flexible framework to do this, by allowing the end user to adjust the number of ancillae used in the integrated gradients calculation. Decreasing the number of ancillae used will decrease the compilation and error correction overhead, at the cost of needing additional circuit executions and runtime. We have revised the discussion of this tradeoff to further emphasize its utility in the EFTQC era, in section 4.3
>
> We sincerely appreciate your feedback and hope that our revisions have addressed your concerns. We would be grateful for any further comments or suggestions.

---

> > ### Comment · Reviewer_Nf8r · 2025-11-26
> >
> > I thank the authors for the answers. I do not have further questions.

---

> > > ### Author Response · Authors · 2025-11-26
> > >
> > > Thank you, Reviewer Nf8r. May we request you update your score as appropriate if your questions have been clarified and you have no further questions.

---

### Official Review · Reviewer_iYfG · 2025-10-30

**Soundness:** 3
**Presentation:** 3
**Contribution:** 2
**Rating:** 4
**Confidence:** 4

**Summary:**

The paper proposes HATTRIQ, a general-purpose framework for computing amplitude-based input attribution scores in circuit-based QML models. The approach adapts the integrated gradients (IG) method, used in classical deep learning, to quantum models by leveraging Hadamard test constructions and parameter-shift rules to estimate gradients directly on quantum hardware.

HATTRIQ supports amplitude embedding, a popular encoding strategy in QML, and the authors demonstrate its use in binary classification tasks on benchmark datasets such as Bars & Stripes, MNIST, and Fashion-MNIST. They also introduce a gradient-parallelization strategy to accelerate computation and report visualizations of pixel-level attributions for the tested datasets.

**Strengths:**

Relevance and motivation: The paper addresses a timely problem, explainability in quantum ML models, which is increasingly important as QML gains traction across platforms.

Methodological soundness: The derivations using Hadamard tests and parameter-shift rules are well-founded and clearly described.

Implementation details and scope: The authors validate their approach on multiple datasets and provide extensive visual examples, showing an effort to bridge theory and practice.

Parallelization component: The inclusion of a gradient parallelization scheme makes the approach more computationally feasible on current simulators or small hardware setups.

The authors provide mathematically valid constructions for gradient estimation using Hadamard tests, and the implementation is commendable. However, interpretability in quantum systems requires more than visual gradient maps—it demands explanations that connect attribution scores to physical quantities (e.g., qubit sensitivity, entanglement contribution, or measurement basis relevance). These links are missing. As presented, HATTRIQ shows that integrated gradients can be computed, but not that they are useful.

**Weaknesses:**

Unclear interpretability value: The main issue lies in the semantic meaning of the attribution results. While the paper technically computes integrated gradients on quantum amplitudes, it is never made clear what positive or negative attributions signify in the quantum context. In the classical setting, the sign and magnitude of IG scores correspond to each input’s contribution toward or against a class prediction. However, for amplitude-based encodings, the physical interpretation of these signed contributions is not explained or demonstrated convincingly.
Lack of interpretive consistency across examples: In Figures 1–3, there is no clear correspondence between the sign and the spatial pattern of the attributions and visually salient regions in the input data. For example, in MNIST digits, one could expect attributions to align with stroke regions or empty background, but this pattern is inconsistent. Similarly, for Fashion-MNIST, attributions do not correspond to intuitive parts of the object (e.g., bag body, handles, or trouser legs). As a result, it remains unclear what a practitioner can learn from these attributions.
Currently, it is impossible to judge whether the quantum attributions are faithful or simply artifacts of circuit sensitivity.

Conceptual disconnect: The paper demonstrates how to compute IGs for quantum models, but not why this is useful or what new insight it offers. Without a clearer discussion of the interpretability objective—e.g., identifying relevant qubits, detecting encoding bias, or assessing circuit sensitivity—the proposed method risks being a purely formal adaptation of a classical technique.

Missing discussion of computational cost and noise sensitivity:
The paper does not analyze the measurement cost required to compute integrated gradients for different circuit sizes. Since IG involves multiple interpolated evaluations per input dimension, the total number of circuit executions could scale rapidly. Moreover, there is no discussion on how hardware noise or shot fluctuations affect the attribution stability. Given that the authors emphasize NISQ feasibility, a quantitative noise analysis or measurement budget estimation is essential to assess practical viability.

While the paper is technically correct and well written, it falls short in demonstrating the practical or conceptual value of amplitude-based integrated gradients for interpretability. The adaptation of a classical method to the quantum domain is interesting but insufficiently motivated and only superficially evaluated. Without a clear explanation of what the computed attributions mean or how they enhance understanding of QML models, the contribution remains limited in impact.

**Questions:**

A clearer explanation of what is being attributed—probability amplitude, measurement outcome probability, or circuit parameter sensitivity—would greatly help the reader.

Quantitative comparisons with classical IG attributions for hybrid models (e.g., classical-quantum classifiers) could strengthen the validation.

---

> ### Author Response · Authors · 2025-11-21
>
> Dear Reviewer iYfg,
>
> We appreciate your careful review and thoughtful suggestions. In the following, we summarize the revisions based on your feedback and offer additional clarification on the utility of our approach.
>
>
> **Clarification on Positive vs. Negative Attributions**
>
> We would like to clarify that positive and negative attributions refer to an input’s contribution toward the model prediction, just as they do in the deep learning setting. Our model output is
> $F(x;\theta) = \langle x | U^\dagger(\theta) O U(\theta) | x \rangle$,
> optionally passed through a scalar nonlinearity. With amplitude encoding
> $|x\rangle = \sum_k x_k |b_k\rangle$, Lemma 3.1 shows $\displaystyle \frac{\partial F}{\partial x_k} = 2,\mathrm{Re}\left[\langle b_k | U^\dagger O U | x \rangle\right]$. This model prediction also corresponds to the expected measurement outcome of the physical system, as this result is what is used as the prediction value.
> Thus we attribute how redistributing amplitude (and hence probability mass via the Born rule) onto basis configuration $|b_k\rangle$ changes the observable $O$ along the input path. Positive IG means increasing $x_k$ increases the class score; negative IG means the opposite.
>
>
> We have adjusted the text in section 5.3 to clarify this.
>
> **Usefulness and conceptual contribution**
>
> Our goal is input-level sensitivity analysis for amplitude-encoded data, which classical IG cannot provide on quantum hardware because the dependence on $x$ appears only through $|x\rangle$. The main result, $\partial F / \partial x_k = 2\mathrm{Re}[\langle b_k | U^\dagger O U | x \rangle]$, combined with the Hadamard test, yields a hardware-compatible way to access these derivatives without knowing the internal quantum state of the the computer.
>
> Practically, this enables: (i) checking that a PQC uses semantically meaningful regions (digits strokes, bag straps), (ii) comparing encodings with similar accuracy but different attribution patterns, and (ii) revealing bias toward background or artifacts even when test accuracy is high. We have stated these interpretability objectives more clearly in the introduction.
>
> **Faithfulness of Attributions**
>
> We appreciate the concern and will clarify the patterns in the figures. For NIST and MNIST, attributions consistently concentrate on foreground strokes: in the $3$ vs. $4$ task, pixels forming the “$3$-loop” receive negative scores while the characteristic “$4$” stroke receives positive scores, aligning with the decision boundary.
>
> For Fashion-MNIST, bag vs. sandal attributions emphasize straps and handles; coat vs.\sandal attributions give negative scores to upper-image regions where sandals never appear. We agree that this is not always obvious. In the revision we have shown positive and negative attributions separately to make this more clear
>
> The quantum attributions are faithful, as they are mathematically equivalent to a conventional IG calculation, upon using enough measurements to accurately estimate the expectation values of the Hadamard test. During testing, this was verified by using a finite difference approximation for the gradient, as suggested in the original integrated gradients paper. However, this finite difference is not viable for real quantum applications, so we omitted it from the final version to focus on our hardware native implementation.
>
>
> **Shot fluctuation**
>
> We agree that managing the circuit evaluation budget is critical for the practicality of quantum techniques, especially due to the need for repeat evaluations during computation of the integral. This motivated us to analyze the efficacy of the technique under different shot budgets, as was shown in figure 5. HattriQ is very accurate with only a few shots, highlighting its resource efficiency.
>
>
> **Comparison with Hybrid Models**
> We agree that hybrid quantum-classical models would be an interesting case study for attribution scores. The reason we did not explore it is that computing IG for classical models is straightforward when compared to the quantum case, and adding classical layers before or after the quantum model does not substantially change the methodology. In fully quantum models, classical IG is not directly applicable because $x$ influences the output only through the quantum state. HattriQ fills precisely this gap by providing $\partial F / \partial x_k$ on hardware, independent of the implementation of $V(x)$.
>
> In hybrid architectures, a quantum layer with amplitude encoding can be treated as a differentiable block: HattriQ supplies its input gradient, which can then be combined with classical IG in preceding layers via the chain rule. Due to space limits we did not include such experiments, but we have added a discussion outlining this integration in section 2.2
>
> We are grateful for your insightful comments and constructive feedback, which have significantly enhanced the quality and presentation of this work.

---

### Official Review · Reviewer_E8j1 · 2025-10-31

**Soundness:** 2
**Presentation:** 2
**Contribution:** 2
**Rating:** 4
**Confidence:** 4

**Summary:**

The authors' core contribution is an adaptation of the classical Integrated Gradients (IG) method to the quantum setting. The method is built on two key technical pillars:

A formal derivation (Lemma 3.1) for the gradient of a QML model's expectation value with respect to the individual (real or complex) amplitudes of the input state. A hardware-native circuit construction (Theorem 4.2) that uses the Hadamard test to compute these exact gradients on quantum hardware , thereby avoiding measurement collapse and the exponential cost of classical simulation.

The paper also proposes a multi-ancilla parallelization technique to accelerate the computation of gradient components. The HATTRIQ framework is validated on several image classification datasets (Bars and Stripes, NIST, MNIST, FashionMNIST), demonstrating high-fidelity attribution maps that highlight salient input features.

**Strengths:**

The demonstration that the method is robust to low shot noise (Fig. 5)  is a major practical strength. It suggests that HATTRIQ could be viable even on near-term devices where measurement budgets are constrained.

**Weaknesses:**

- The paper's primary evaluation relies on "ideal error-corrected hardware" and only analyzes the impact of shot noise (statistical sampling error). The proposed Hadamard test circuit (Sec 4.2) requires implementing deep, controlled unitaries, including the entire model $C-(U^\dagger O U)$. This type of circuit is known to be extremely sensitive to hardware/gate noise and decoherence, a much bigger challenge on NISQ-era devices.

- The paper acknowledges that "generic controlled operations can incur additional circuit overhead" but does not analyze this critical bottleneck. The gate complexity of compiling $C-V(x)$ and particularly $C-(U^\dagger O U)$ will be the true scaling limit of this method. A discussion of the compilation overhead (e.g., in terms of CNOT count) is necessary for a complete picture of the method's "scalability."

- The attribution results are contingent on the choice of a "blank image (0 for all pixel values)" as the baseline $x'$. In classical IG, the baseline choice is a known, sensitive parameter. A zero-vector is an "off-manifold" point that may not be a meaningful reference. The paper would be strengthened by a discussion of this choice or a brief exploration of alternative baselines (e.g., an average image from the dataset)

**Questions:**

1. Could the authors provide a more concrete analysis of the gate complexity overhead required to implement the controlled unitaries, specifically $C-V(x)$ and $C-(U^\dagger O U)$? This seems to be the primary scalability bottleneck.

2. Can the authors comment on the expected robustness of HATTRIQ to hardware/gate noise? The proposed Hadamard test circuit seems particularly vulnerable to decoherence, which could severely degrade the fidelity of the inner product estimation.

3. The choice of a zero-vector baseline  is simple, but often problematic in classical IG. Have the authors considered other baselines (e.g., averaged images, blurred images) and how this might affect the attribution maps?

---

> ### Author Response · Authors · 2025-11-21
>
> Dear Reviewer E8j1
>
> Thank you for your constructive feedback. Below, we highlight the changes we implemented in response and provide some clarifications concerning the overhead and baseline choice of our technique.
>
> **Hardware Sensitivity**
>
> We agree that the implementation of deep, controlled unitaries is sensitive to hardware noise, which is why we choose to focus on devices featuring some level of fault tolerance, where hardware error is less impactful than it is in the NISQ era.
>
> **Compilation Overhead**
>
> $C-V(x)$ for the computational basis states is very simple, requiring only a linear (in qubit count) number of CNOT gates from the ancilla to the data register.
>
> $C-V(x)$ for the general amplitude encoding circuit is difficult to quantify, owing to the variable number of gates present in a general purpose amplitude encoding circuit. It is likely that this overhead could be cut down by using approximate state preparation methods   such https://doi.org/10.1103/PhysRevA.109.042401 or https://doi.org/10.48550/arXiv.2503.14473, or if the data has some specific structure known in advance.
>
> C-$U^\dagger O U$ The key feature of our technique is that gradients are computed from expectation values. While this still requires a controlled version of the quantum model, C-$U^\dagger O U$, it is a direct expectation value measurement, which avoids the resource-intensive process of hidden-state tomography or classical simulation.
>
> In essence, the difficulty in implementing the technique centers around C-$U^\dagger O U$, which, for a given architecture, might be designed/chosen so that it may be more efficiently compiled or approximated than general multi-controlled gates, especially in hardware that supports native multi-qubit interactions. While it is true that multicontrol gates do incur high overhead on superconducting hardware, it is possible that this overhead might be avoidable on other hardware platforms. One example is [neutral atoms](https://doi.org/10.1103/PhysRevLett.123.170503), which feature tunable qubit interactions determined by the spatial configuration of atoms within the computer. A second example is [trapped ions](https://doi.org/10.1103/PhysRevA.84.012314), which have a high degree of connectivity owing to the ability to utilize global “motional” states that can entangle subsets of qubits within the computer.
>
> For demonstration purposes, we focus on layers of CNOTs within a parameterized quantum circuit due to their prevalence in the literature, however, it is entirely possible to substitute this circuit for another type of quantum model, based on these native hardware constraints.
>
> We have added a more lengthy discussion of this to section 4.2
>
> **Baseline Choice**
>
> We did try alternative baselines during the testing phase of HattriQ. Ultimately, we settled on blank images, due to their simplicity and the visual clarity of the attributions produced. We did also try using an average image, but initially omitted it in order to focus on our circuit design contributions. We have added the results for the MNIST dataset to the revised appendix G.
>
> We are truly grateful for your insights and hope that the changes we’ve made resolve your concerns. Please let us know if further clarification is needed.

---

### Official Review · Reviewer_iYGU · 2025-10-31

**Soundness:** 3
**Presentation:** 3
**Contribution:** 2
**Rating:** 0
**Confidence:** 5

**Summary:**

The paper introduces a new interpretability technique for a subclass of quantum machine learning (QML) methods. The interpretability technique, called HattriQ, is designed for QML models that use parametrized quantum circuits. HattriQ computes attribution scores for input features by extending the populare integrated gradient framework from classical ML. The paper's contributions are as follows:
1. HattriQ, the aforementioned interpretability technique.
2. A simple-to-implement technique for calculating the gradients used by HattriQ on quantum hardware.
3. A proposal for computing attribution scores in parallel on large quantum processors.
4. Several demonstrations on simulated noise-free quantum processors.

**Strengths:**

1. Interesting Technique: The paper presents a novel method (HattriQ) for computing input attribution scores for quantum circuits, which is an intriguing approach in the field of quantum machine learning (QML).
2. Theoretically Sound Method: The method is theoretically sound and extends a popular classical technique, which adds value to the existing body of knowledge.
3. Demonstrations on Classic ML Datasets: The paper includes several demonstrations on well-known classical machine learning datasets (e.g., Bars and Starts, MNIST, FashionMNIST), showcasing the applicability of the method, at least on error-free quantum processors.
4. Growing Field: The focus on interpretable QML aligns with the increasing interest in making AI models more understandable, which is crucial for broader adoption.
5. Acknowledgment of Challenges: The paper correctly identifies the lack of interpretability methods as a barrier to AI adoption, which is an important consideration for the future of QML.

**Weaknesses:**

Since the paper performed demonstrations on simulated noise-free hardware, I will focus my analysis on HattriQ's relevance to fault-tolerant quantum computing (FTQC).

**Major weaknesses**
1. Scalability Challenges: The technique requires as many ancilla qubits as there are features, negating the space savings typically associated with QML.
2. Implementation Challenges: The method necessitates complicated controlled gates, with no clear guidance on how to implement these gates on FTQC systems or reckoning with the overhead.
3. Limited Long-Term Utility: The paper fails to reckon with QML's uncertain future in FTQC. At the moment, QML is not a serious application for FTQC without significant technical advancements (due to overhead, difficulties with loading classical data, lack of realizable advantages, and barren plateaus), which raises concerns about its practical relevance.

**Minor weaknesses**
1. Inadequate Support for Some Claims: The references in the introduction do not adequately support the claims made, particularly regarding the potential speedups of QML algorithms over classical methods.
2. Insufficient Background Information: The quantum states and gates background provided may not adequately prepare readers unfamiliar with QML, lacking explanations of key concepts like observables and parameterization of gates.
3. Unclear Accuracy Metrics: It is unclear what the "accuracy scores" in the paper refer to. Are they the accuracy scores of the QML models or of HattriQ?

**Questions:**

How robust is HattriQ to device noise?

Were the trained QML classifiers any good?

Is there a way around using complicated controlled gates to compute the input feature gradients?

---

> ### Author Response · Authors · 2025-11-21
>
> Dear Reviewer iYGU,
>
> Thank you for your comments and insights. Below we discuss some of the revisions we made based on your feedback, and offer some clarifications surrounding the utility of our technique.
>
> **Scalability / Implementation Challenges**
>
> The key feature of our technique is that gradients are computed from expectation values. While this still requires a controlled version of the quantum model, C-$U^\dagger O U$, it is a direct expectation value measurement, which avoids the resource-intensive process of hidden-state tomography or classical simulation.
>
> In essence, the difficulty in implementing the technique centers around C-$U^\dagger O U$, which, for a given architecture, might be designed/chosen so that it may be more efficiently compiled or approximated than general multi-controlled gates, especially in hardware that supports native multi-qubit interactions. While it is true that multicontrol gates do incur high overhead on superconducting hardware, it is possible that this overhead might be avoidable on other hardware platforms. One example is [neutral atoms](https://doi.org/10.1103/PhysRevLett.123.170503), which feature tunable qubit interactions determined by the spatial configuration of atoms within the computer. A second example is [trapped ions](https://doi.org/10.1103/PhysRevA.84.012314), which have a high degree of connectivity owing to the ability to utilize global “motional” states that can entangle subsets of qubits within the computer.
>
> For demonstration purposes, we focus on layers of CNOTs within a parameterized quantum circuit due to their prevalence in the literature, however, it is entirely possible to substitute this circuit for another type of quantum model, based on these native hardware constraints.
>
> We have added a more lengthy discussion of this to section 4.2
>
> **Ancillae Requirements**
>
> We would like to emphasize that the number of features attributed scales exponentially with the number of ancillae, not linearly, so that the space savings of the quantum model are not negated. To see this, note that in the circuit diagram of section 4.3 each ancilla bitstring corresponds to one prepared basis state, with the exception of the “11” string, which is used to prepare the data. The number of features extracted scales with the number of ancilla basis states; that is exponentially with the number of ancilla qubits. We have revised part of section 4.3 to emphasize this point and provided a more formal derivation in Appendix D.
>
> **Avoiding the Controlled Unitaries**
>
> In general, we would not expect to be able to remove the controlled unitaries from HattriQ, as our Hadamard test relies on the phase difference between the various prepared states in the data register. For a longer discussion, we refer to [this paper](https://doi.org/10.48550/arXiv.2508.00055), especially footnote 3. As noted there by Tang and Wright, the controlled unitaries in the Hadamard test can be removed when the end goal is to estimate the absolute value $|\bra{\psi}\phi \rangle|$. However, for our application, we explicitly need the real part of the inner product, Re${\bra{\psi}\phi \rangle}$, making the controlled unitaries non-removable.
>
> **Long-Term Utility**
>
> Importantly, our technique is independent of the model architecture used, and not just restricted to Variational Quantum Algorithms (VQAs) / Parameterized Quantum Circuits. While we do focus on VQAs as a case study of the technique, it is entirely possible to exchange the variational model with another kind of unitary circuit, making HattriQ adaptable to future innovations in Quantum Machine Learning (QML) research. We would also like to highlight there is progress being made in translating VQAs to the fault tolerant era, and many [open questions](https://arxiv.org/pdf/2501.05694) remain as to what limitations are truly fundamental.
>
> A common theme of Early Fault Tolerant Quantum Computing is to [trade reduced depth/error correction overhead for increased sampling complexity/runtime ](https://doi.org/10.1103/PhysRevResearch.6.023118). HattriQ provides a natural and flexible framework to do this, by allowing the end user to adjust the number of ancillae used in the integrated gradients calculation. Decreasing the number of ancillae used will decrease the compilation and error correction overhead, at the cost of needing additional circuit executions and runtime. We have revised the discussion of this tradeoff to further emphasize its utility in the EFTQC era, in section 4.3
>
> Finally, while the future surrounding QML is still unclear, we would argue that interpretability techniques could play a role in uncovering the viability of QML, and may help identify what limitations do exist in the quantum learning process.
>
> **Support of Claims in Introduction**
> We have added additional references to the introduction that further support our claims.

---

> > ### Author Response · Authors · 2025-11-21
> >
> > Continued from above:
> >
> > **Background Information**
> >
> > Additional background information has been provided in appendix A
> >
> > **Unclear Accuracy Metrics**
> >
> > The accuracy scores reported in the table are accuracies for the trained classifier models, not HattriQ. We revised the table’s caption to reflect this.
> >
> > **Device Noise**
> >
> > One challenge in quantifying the effects of hardware noise is that it is dependent on both the hardware and also the error correction codes utilized, with specific details such as T1 and T2 times, gate error rates, encoding rates, and code thresholds all drastically changing the noise profile. As it is still unclear which correction scheme/hardware combination will achieve mass adoption, we omitted noisy simulations to avoid overgeneralizing our results, as any model we create would be a conditional statement about performance, not a general statement about the algorithm's potential.
> >
> > We would like to thank the reviewer for their detailed evaluation and valuable recommendations, which have strengthened both the rigor and readability of our paper. We hope the revisions adequately address your concerns and would welcome any additional guidance.

---

> ### Comment · Reviewer_iYGU · 2025-11-21
>
> Thank you for your response. I appreciate the updated clarity on the ancillae requirements. In light of this clarification, I believe I was too harsh in my initial assessment and will increase my grade slight. However, I still have strong reservations about the real-world applicability of HattriQ, in the early fault-tolerant era and beyond.
>
> "In essence, the difficulty in implementing the technique centers around C-$U^\dagger O U$, which, for a given architecture, might be designed/chosen so that it may be more efficiently compiled or approximated than general multi-controlled gates, especially in hardware that supports native multi-qubit interactions. While it is true that multicontrol gates do incur high overhead on superconducting hardware, it is possible that this overhead might be avoidable on other hardware platforms. One example is neutral atoms, which feature tunable qubit interactions determined by the spatial configuration of atoms within the computer. A second example is trapped ions, which have a high degree of connectivity owing to the ability to utilize global “motional” states that can entangle subsets of qubits within the computer."
>
> I don't believe that this paragraph accurately reflects the feasibility of implementing fault-tolerant multi-control gates on logical qubits in atom-based quantum computers. Multi-qubit logical controlled gates will almost certainly remain costly for the foreseeable future. To start, while the all-to-all connectivity of atom-based quantum computers opens the door to exotic fault-tolerant architectures with "easy" logical multi-controlled gates, I am unaware of any existing architectures for which c^n-U (i.e., n-qubit controlled-U for arbitrary U and n) are "easy." As a result, implementing these gates will almost certainly require the use of magic states and incur substantial overhead.
>
> In general, the most promising path to leveraging atom-based platforms for reducing FT overheads is through the use of qLDPC codes. But there is not a clear connection between using qLDPC codes and reducing the overhead of multi-qubit controlled gates beyond the general overhead savings from using qLDPC codes. Most importantly, it still seems unlikely that the use of qLDPC codes will make QML a viable application for large-scale fault-tolerant quantum computers. The resource requirements for FT implementations of QML algorithms with provable advantage are astronomical and the evidence for heuristic advantage on meaningful problems are pretty scant.

---

> > ### Author Response · Authors · 2025-11-21
> >
> > Dear Reviewer iYGU,
> >
> > Thank you again for your prompt follow-up and for increasing your score. We genuinely appreciate the time and care you have invested in evaluating our work.
> >
> > We hope you will allow us to raise one broader question about the principle underlying your latest concern. Your commentary suggests that research directions should be considered relevant only if (1) current theory already provides proof that they will matter in the fully fault-tolerant (FT) era, and/or (2) current hardware already supports the required primitives at scale. While quantum theory and quantum hardware research are undeniably important and impactful, we respectfully ask whether this is truly the intended position, because if interpreted literally, this criterion would significantly narrow the scope of quantum algorithm and QML research. Historically, scientific progress has not operated in this manner. Humanity built buildings, bridges, and steam engines long before the mathematical or physical theories explaining them existed. Practice informed theory instead of waiting for theory to dictate practice.
> >
> > This is precisely the viewpoint articulated repeatedly by John Preskill, arguably the most influential figure currently in quantum computing, as recently as November 2025. For instance, [Eisert and Preskill (2025)](https://arxiv.org/pdf/2510.19928) write that early applications of quantum computing "will be primarily scientific" and that "scientific exploration enabled by near-term quantum computers will form the foundation for a variety of unforeseen applications in the longer term." They emphasize that hardware modalities that were previously dismissed (such as Neutral arrays) have, in just a few years, become leading candidates for early logical-qubit demonstrations. Their conclusion could not be clearer: "We do not know yet which quantum computing modalities will be best suited," and therefore we must continue to explore the idea. The article ends on this note: "Despite our best efforts to predict the important applications, tomorrow's quantum computers are sure to delight and benefit us in ways we cannot currently anticipate. Before that happens, we have a lot of work to do."
> >
> > Likewise, in [Zimborás et al. (2025)](https://arxiv.org/pdf/2501.05694) the authors stress that one must "reevaluate how we think about quantum computing before full fault tolerance is achieved" and caution strongly against allowing negative theoretical results to "lead to a dismissal of real opportunities" because "theoretical no-go results often pertain only to the asymptotic regime" and "should not be misinterpreted as definitive statements about practical performance." Pertaining to QML, they also show evidence to suggest that "While some technical challenges, such as high circuit repetition counts and fine rotation-angle resolution, need more attention, the community is making progress in addressing these, and some form of variational quantum algorithms will likely find useful applications in the fault-tolerant quantum computing era; much like in classical computing where variational methods are very prominent."
> >
> > We raise this not to dispute your technical points about the cost of multi-controlled logical operations (we fully acknowledge these challenges) but to ask whether it is desirable for our community to limit conceptual progress only to those tasks that today's FTQC resource estimates already deem feasible. Such a restriction would mean that QML interpretability, new quantum learning models, hybrid quantum-classical techniques, and architecture-informed algorithm design could not be pursued unless the fully fault-tolerant resource story is already known. The concern we express is that this view would dramatically contract the exploratory space of the field at precisely the time when the field needs broader conceptual experimentation.
> >
> > HattriQ is not designed for a particular FT architecture. It is a conceptual framework that tries to understand how quantum models learn. This aligns precisely with the principle emphasized by recent community perspectives: we cannot predict which algorithmic ideas will become relevant once FT machines exist and technologies mature. Of course, HattriQ can be updated in future iterations to cater to the specific requirements of different FT machines as hardware specifications evolve and are finalized. Thus, this is not a technical limit.
> >
> > Therefore, we respectfully urge reconsideration of the underlying premise that only FT-feasible methods, to date, merit attention. Doing so may inadvertently suppress innovation, particularly in areas such as QML interpretability, where the cost structure of future hardware could differ dramatically from what fault-tolerant resource theories currently assume. Exploratory algorithmic ideas play an essential role in shaping the future landscape.
> >
> > We again thank you for your thoughtful review, for the increase in score, and for engaging so constructively with our work.

---

> > > ### Comment · Reviewer_iYGU · 2025-11-22
> > >
> > > I realize that my earlier comment may have left room for misinterpretation, so I would like to clarify. Exploratory work is, of course, valuable and important. But the key questions for publication are whether a given piece of work fits within the venue and what standards it should meet in order to appear there. As a first-order guideline, research proposing algorithms intended for fault-tolerant hardware should be evaluated in terms of their viability on such hardware; when a clear or well-supported path to fault-tolerant implementation is lacking, the work may be a better fit for more specialized venues. Similarly, supplementary contributions—such as interpretability tools for QML—should be assessed in part by considering the significance of the work they are designed to support. These criteria are not absolute: truly seminal, broadly impactful advances that are likely to interest researchers across disciplines are clear exceptions. Moreover, thresholds change over time.
> > >
> > > While I commend the authors for conducting and presenting solid research, I currently do not see a compelling argument that HattriQ’s reliance on arbitrary many-qubit controlled unitaries makes it suitable for fault-tolerant quantum computation without further resource estimation, especially given the challenges associated with scaling QML to large-scale FTQC. As a result, I don’t believe the work passes question 3.4 in the reviewer guidelines: what is the significance of the work?
> > >
> > > To clarify the kinds of demonstrations that would strengthen the case for significance, I would find the following particularly compelling:
> > >
> > > **Concrete examples of HattriQ applied to techniques with a clearer path to practical advantage** in either the early fault-tolerant or fully fault-tolerant era.
> > >
> > > **Evidence that HattriQ remains robust to noise at scales relevant to early fault-tolerant devices**, taking into account the overhead associated with implementing many-qubit controlled unitaries. unitaries).

---

> > > > ### Author Response · Authors · 2025-11-22
> > > >
> > > > Dear Reviewer iYGU,
> > > >
> > > > Thank you once again for engaging with our work and for continuing the conversation with us. Your latest response has clarified some aspects for us about the root of your concerns. After carefully reviewing your message, it now appears that your concerns fall into two main categories (as indicated by the last two bolded lines in your message). Fortunately, we are glad to address these concerns in a straightforward manner below. You wrote the following:
> > > >
> > > > **(1)** You would like to see *"HattriQ applied to techniques with a clearer path to practical advantage in either the early fault-tolerant or fully fault-tolerant era"* for it to appear at ICLR. And you stated that the reason for this is that *"research proposing algorithms intended for fault-tolerant hardware should be evaluated in terms of their viability on such hardware; when a clear or well-supported path to fault-tolerant implementation is lacking, the work may be a better fit for more specialized venues."*
> > > >
> > > > **Ans:** This is a concern of fit for ICLR and whether ICLR only accepts works on areas with a theoretically proven quantum advantage. We appreciate you raising this fit question, and we are glad to clarify this with evidence. We hope it is helpful to note that ICLR has long been an active and welcoming venue for research on quantum machine learning tasks, including quantum classification and works relevant to fault-tolerant architectures, with no proven quantum advantage to date.
> > > >
> > > > To provide just a few references, [Kerenidis et al. (2020)](https://openreview.net/forum?id=Hygab1rKDS) proposed a quantum convolutional neural network architecture capable of implementing large-scale image classifiers using quantum primitives, explicitly exploring the model’s capabilities in deep learning contexts. More recently, [Gil-Fuster et al. (2025)](https://openreview.net/forum?id=TdqaZbQvdi) tackled the interplay between dequantization and trainability in variational quantum classifiers, with controlled unitary gates in their structure (same as HattriQ, thus similarly targeting fault-tolerant regimes). On the other hand, [Lei et al. (2023)](https://openreview.net/forum?id=8htNAnMSyP) studied when variational classifiers could be enhanced with quantum kernels and offer advantages in practical quantum-classical settings. These works complement broader QML classification studies at ICLR, such as [Fischbacher and Sbaiz (2021)](https://openreview.net/forum?id=CHLhSw9pSw8), which explored classification tasks in optical quantum setups using the MNIST and Fashion-MNIST datasets, similar to HattriQ. Collectively, these contributions highlight a strong and growing precedent within ICLR for quantum machine learning classification research, including frameworks that anticipate or directly address the era of fault tolerance. We hope this helps affirm the appropriateness of ICLR as a venue for our contribution.
> > > >
> > > > **(2)** You would also like to see that *"HattriQ remains robust to noise at scales relevant to early fault-tolerant devices."*
> > > >
> > > > **Ans:** We appreciate this request and would like to clarify a key misunderstanding. By definition, fault-tolerant quantum computers do not exhibit hardware noise at scales that dominate NISQ devices. Fault tolerance specifically suppresses physical noise using error correcting codes, so the computation that the logical algorithm sees is essentially noiseless aside from sampling or shot noise (already explored in Figure 5 in the paper), which is intrinsic to all quantum algorithms. Thus, there is no meaningful notion of demonstrating *"noise scaling"* on fault-tolerant machines as the noise levels are suppressed to the level of abstraction from the logical algorithm. The results are essentially equivalent to those demonstrated in the ideal setting presented in Figures 1-9 in the paper. As a result, there is no concept of hardware noise propagation through HattriQ that requires characterization. What matters is the structural compatibility of the algorithm with fault-tolerant primitives, which we have discussed extensively before and which can be optimized as architectures evolve.
> > > >
> > > > We would like to conclude by reiterating our appreciation for your engagement and the clarity with which you communicated your expectations. We believe these clarifications address your concerns in full, and we appreciate the opportunity to provide them.

---

> > > > > ### Comment · Reviewer_iYGU · 2025-11-22
> > > > >
> > > > > I won’t comment too deeply on papers that I have not read, but I don’t think that any of the cited papers run counter to main arguments in my last comment.
> > > > >
> > > > > Early fault-tolerant quantum computers are precisely those quantum computers which bridge the gap between the NISQ era and the fault-tolerant application-scale quantum era outlined in Eisert and Preskill (2025). As you noted in an earlier comment, this nascent era is characterized by partially or fully fault-tolerant devices that are only able to achieve middling logical error rates (order 10^-4 to say 10^-8), thus necessitating serious considerations around tradeoffs. Can you show how HattriQ performs on such devices?

---

> > > > > > ### Author Response · Authors · 2025-11-22
> > > > > >
> > > > > > Yes, of course. Please allow us a couple of days to run HattriQ in those noise regimes and we’ll update the paper with those results. As devices in those regimes do not exist as of yet, these will of course have to be noise-enabled simulation runs. We’ll comment back once we’ve updated the paper. Thank you for the opportunity.

---

> > > > > > > ### Author Response · Authors · 2025-11-26
> > > > > > >
> > > > > > > Dear Reviewer iYGU,
> > > > > > >
> > > > > > > Thank you again for giving us the opportunity to follow up with the noisy-simulation results you requested. We have now added these experiments to Appendix H, where we evaluate HattriQ under depolarizing noise consistent with early fault-tolerant operating regimes.
> > > > > > >
> > > > > > > As shown in Appendix H (Fig. 10), the attribution maps remain highly stable at an error rate of $10^{-4}$, with minimal deviation from the ideal simulation. We selected $10^{-4}$ because this level will soon reflect the effective noise regime of today's highest-quality devices, and, as you noted, represents a realistic target for early fault-tolerant machines. For stress testing, we also explored significantly higher noise levels (e.g., $10^{-1}$), where attribution quality does degrade, as expected (these results are not currently in the revision, but we’re happy to include them if the reviewer desires). However, since current NISQ devices already operate near $10^{-4}$, and early FTQC will only further suppress errors from this baseline, we, of course, do not anticipate performance worse than what is shown in Appendix H.
> > > > > > >
> > > > > > > We hope these additional results directly address your request and provide clarity on HattriQ's robustness in the noise regimes most relevant to early fault-tolerant computation.

---

### Author Response · Authors · 2025-11-12

Dear Reviewers,

Thank you for your thorough feedback. We have received your comments and are in the process of analyzing them. You can expect a detailed response addressing each point soon.

Best regards,
Authors

---

### Author Response · Authors · 2025-11-21

Thank you to all the reviewers for your helpful comments and questions. We sincerely appreciate your efforts in providing us with feedback, and we believe that incorporating this discussion has improved the quality of this work. We have revised parts of the text (highlighted in blue), and added an additional figure in Appendix G based on your feedback. We also revised the parallel gradient derivation (appendix D), making it more explicit to better reflect the exponential scaling of input features with respect to ancilla qubit count. We have also separated the positive attributions from the negative attributions in figures 2 and 3, to make the patterns more clear.

We have provided additional context for these changes as direct responses under each of your original comments, and have also replied to specific concerns each of you brought up. Please reach out with any additional questions or concerns.

Thank you,

The Authors

---

### Author Response · Authors · 2025-11-29
**Summary of Our Revisions and Clarifications**

Dear Area Chairs,

We are deeply grateful to the reviewers and the program committee for the exceptional care invested in evaluating our manuscript. The feedback was invaluable and substantially improved both the technical rigor and clarity of the paper. Here, we provide a consolidated summary to directly support the AC’s evaluation of how all raised concerns were comprehensively addressed.

**Scalability and ancilla requirements.** We clarified the scaling behavior in Sec. 4.3 and provided a formal derivation in Appendix D, showing that the number of attributable features scales exponentially with the number of ancilla qubits, rather than linearly, thereby preserving the space efficiency of amplitude encoding. We explicitly framed the parallelization strategy as a tunable tradeoff between ancilla count and sampling cost, consistent with early fault-tolerant execution models.

**Controlled-unitary overhead and hardware feasibility.** In Sec. 4.2, we expanded our discussion of circuit compilation and architectural adaptability, emphasizing that HattriQ is an expectation-value-based framework that avoids state tomography or classical simulation. We made explicit that model circuits can be selected to reflect native gate sets on different platforms and that the CNOT-based ansätze were used as representative examples rather than rigid requirements. We also acknowledge the multi-controlled Toffoli gate costs, while clarifying that the framework remains architecture-agnostic and is designed to benefit from maturing fault-tolerant compilation strategies.

**Noise robustness in early fault-tolerant regimes.** At the reviewer’s request, we added new noise-enabled simulations in Appendix H. These experiments evaluate depolarizing noise at target logical error rates of $10^{-4}$ representative of early fault-tolerant regimes and demonstrate that attribution maps remain stable with minimal deviation from ideal results. Additional stress tests at higher noise levels confirm the expected degradation behavior, validating the practical robustness of HattriQ as intended for operation.

**Shot noise and sampling efficiency.** In Sec. 5.4 and Fig. 5, we introduced a detailed shot-budget analysis showing that accurate attribution quality is achieved with as few as tens of shots per gradient component. These results establish the measurement efficiency of HattriQ under realistic execution budgets.

**Faithfulness and semantic interpretation of attributions.** We clarified this core point in Sec. 5.3 by explicitly linking positive and negative integrated gradients to the physical prediction function $F(x;\theta)=\langle x|U^\dagger(\theta)OU(\theta)|x\rangle$. We explain how redistributing amplitude via the Born rule leads to signed contributions toward or against class scores exactly analogous to classical IG semantics. To enhance visual interpretability, we have separated the positive and negative attribution maps in Fig. 2 and 3, and expanded the accompanying analysis to demonstrate alignment with semantically meaningful features across the MNIST, NIST, and FashionMNIST tasks.

**Baseline sensitivity.** We evaluated alternative baselines and added comparative experiments using dataset-average baselines to Appendix G, while retaining the blank baseline for clarity. We document that the qualitative attribution patterns remain consistent across baseline choices.

**Background accessibility.** To better support non-specialist readers, we expanded explanatory material on quantum observables, parametrized circuits, and measurement-based prediction in Appendix A.

**Accuracy metrics clarification.** We revised the text to state unambiguously that reported accuracy values correspond to classifier performance rather than attribution accuracy.

**Literature support and framing.** We strengthened the Introduction and Related Work sections by adding updated citations and more carefully positioning our contribution within the context of active QML research and early fault-tolerant exploration. We clarified that interpretability tools, such as HattriQ, serve as foundational analysis instruments for evaluating and guiding quantum learning models.

In summary, we believe we have methodically and fully addressed every reviewer concern by adding formal derivations (Appendix D), expanding background exposition (Appendix A), refining hardware and overhead discussion (Sec. 4.2 and 4.3), introducing early fault-tolerant noise robustness experiments (Appendix H), analyzing shot efficiency (Sec. 5.4), strengthening attribution faithfulness explanations (Sec. 5.3 and Fig. 2–3), exploring alternative baselines (Appendix G), and clarifying evaluation metrics throughout. The reviewers’ feedback was instrumental in driving these improvements, and we are grateful for the opportunity to revise. We respectfully submit that the revised paper now presents a comprehensive, rigorous, and practical contribution that fully addresses all the concerns raised.

---

### Meta-Review · Area_Chair_ePRr · 2026-01-07

**Summary:**

All reviewers raised serious concerns, mainly about the practicability. As all recommendations are consistent, I'm inclined to reject this paper.

**Reviewer Scores:**

NA

---

### Decision · Program_Chairs · 2026-01-26

Reject